psychology/neuroscience/behaviour

replication success, *p*-values, meta-analysis, Bayes factors, publication bias

**Author for correspondence:**
Jasmine Muradchanian
e-mail: jasmine.muradchanian@gmail.com

# How best to quantify replication success? A simulation study on the comparison of replication success metrics

Jasmine Muradchanian, Rink Hoekstra, Henk Kiers and Don van Ravenzwaaij

Behavioural and Social Sciences, University of Groningen, The Netherlands

JM, 0000-0002-2914-9197; RH, 0000-0002-1588-7527; HK, 0000-0002-4995-9349; DvR, 0000-0002-5030-4091

To overcome the frequently debated crisis of confidence, replicating studies is becoming increasingly more common. Multiple frequentist and Bayesian measures have been proposed to evaluate whether a replication is successful, but little is known about which method best captures replication success. This study is one of the first attempts to compare a number of quantitative measures of replication success with respect to their ability to draw the correct inference when the underlying truth is known, while taking publication bias into account. Our results show that Bayesian metrics seem to slightly outperform frequentist metrics across the board. Generally, meta-analytic approaches seem to slightly outperform metrics that evaluate single studies, except in the scenario of extreme publication bias, where this pattern reverses.

## 1. Introduction

In recent years, the quality of published literature has been questioned in various scientific disciplines such as psychology (e.g. [1–4]), economics [5,6] and medicine ([7–10]; see also [11] for a discussion in neuroscience). To overcome this crisis of confidence, replicating original findings is becoming increasingly more common in the scientific community (e.g. [12–18]).

Despite the increasing prevalence of and attention for replication, there are few clear-cut definitions of and little agreement on what constitutes a successful replication. The focus of the present study is on close replications, which 'aim to

recreate a study as closely as possible, so that ideally the only differences between the two are the inevitable ones (e.g. different participants)' [19, p. 218]. Rather than exact or direct replications, we prefer the term 'close replications' because it highlights that 'there is no such thing as an exact replication' [20, p. 92]. In the present study, we make an attempt to answer '… the question of how to judge and quantify replication success' [21, p. 1457].

A variety of methods, both frequentist and Bayesian, has been proposed to quantify replication success. Some of them have been widely used in various large replication projects across scientific fields such as psychology [18], economics [12], social sciences in general [13] and experimental philosophy [15]. So far, there does not seem to be a consensus on a single golden standard to quantify replication success, which explains why typically several metrics are reported [12,13,15,18]. As a result, whether or not a replication attempt is classified as being successful can depend on the type of measure being used. For example, using the traditional significance method, Open Science Collaboration (OSC, [18]) concluded that 36% of the replication studies in their sample were successful, whereas when combining the original and replication effect sizes for cumulative evidence by using fixed-effects meta-analyses, the number of significant effects was 68%.

With a plethora of available quantitative measures of replication success, it might be difficult for replication researchers to decide which indicator(s) they should use to quantify replication success. Therefore, it is important to gain more insight into how these metrics function.

In one of the few studies that examines features of methods of quantifying replication success, Schauer & Hedges [22] focused on examining the quantitative properties of three different replication success methods: the significance criterion, the confidence interval overlap criterion and the prediction interval. They also provide a qualitative discussion of meta-analytic averaging. In their study, they analytically assess two types of classification errors: (1) false success rate, which is the proportion of times 'the analysis concludes that the studies successfully replicated when they did not (for a given definition of replication)' [22, p. 3] over all times the analysis concludes that the studies successfully replicated and (2) false failure rate, which is the proportion of times of an 'analysis concluding that the studies failed to replicate when they actually successfully replicated according to a given definition' [22, p. 3] over all times the analysis concludes that the studies failed to replicate. In this work of Schauer and Hedges, the definition of replication success is test-centric in the sense that a successful replication involves compatible test results. Compatible can mean that effect sizes are in a certain range from one another (Definition 1) or have the same sign (Definition 2).

A complicating factor in the comparison of metrics for quantifying replication success is publication bias, which means that studies reporting significant findings have a higher probability to be published than studies reporting non-significant findings (e.g. [23,24]). Publication bias has probably persisted because journals mostly seem to accept studies that are novel (i.e. they prefer original studies over replications; e.g. [25]), good (i.e. 'clear, coherent, engaging, exciting'; e.g. [26, p. 204]) and statistically significant (usually, $p < 0.05$; e.g. [27]). On the other hand, publication bias has probably also persisted because researchers are more likely to submit significant than non-significant results for publication [28]. Given the fact that presenting significant results enhances the probability of a paper getting published [29], researchers often deviate from their original designs (by, for example, adding observations, dropping conditions, including control variables, etc.), sometimes without being aware that this artificially inflates Type I error rates [4]. Thus, the higher rejection rate for non-significant findings and the pre-emptive response by authors to let their methods and analyses depend on the significance of their findings, both result in a bias of reporting sample effect sizes that probably overestimate the true effect size (e.g. [18]). This bias will have repercussions for replication success (e.g. the rate of replication success will be lower; [3]). However, it is unclear to what extent publication bias affects different metrics of replication success.

In the present study, we aim to evaluate and compare a number of quantitative measures of replication success. Schauer & Hedges [22] restricted themselves to the significance criterion, the confidence interval overlap criterion, and the prediction interval, presumably because results for these can be calculated analytically. The authors did not examine other (popular) metrics for quantifying replication success, such as the Bayes factor (BF), Bayesian meta-analysis, the Small Telescopes method and the sceptical *p*-value, for which no analytical solutions are readily available. In our paper, we therefore resort to simulation to examine the properties of these metrics. In addition, contrary to Schauer & Hedges [22], we explicitly compare metrics against one another in terms of their classification accuracy. We employ a different definition of classification accuracy from Schauer & Hedges [22] that is not concerned with compatible test results, but instead focus on whether methods *correctly* conclude there is an effect when there is *one* and methods *incorrectly* conclude there is an

**Table 1.** Replication success criteria for each quantitative measure of replication success. *Note*: we conducted one-sided one-sample *t*-tests for all studies (i.e. $H_0$: $\mu = 0$ and $H_a$: $\mu > 0$). Therefore, we were only interested in finding positive effect sizes. The alpha levels range from 0.0001 to 0.5 in increments of 0.0001 (i.e. 5000 thresholds); the BF thresholds range from 1/10 to 200 in increments of 1/25 (i.e. 4998 thresholds).

| metric | replication success criteria |
| --- | --- |
| (1) significance | original and replication studies have a positive effect size, in addition to replication study being significant; we included a range of alpha levels from 0.0001 to 0.5 |
| (2) Small Telescopes | replication effect size is not significantly smaller than an effect size that would have given the original study with a positive effect size a statistical power level of 33%; we used 0.05 as level for alpha |
| (3) classical meta-analysis | original study and fixed-effect meta-analysis have a positive effect size, in addition to meta-analysis being significant; we included a range of alpha levels from 0.0001 to 0.5 |
| (4) BF | original and replication studies have a positive effect size, in addition to the JZS BF of the replication study being larger than a BF threshold; we included a range of BF thresholds from 1/10 to 200 |
| (5) replication BF | the replication BF is larger than a BF threshold, given that the original study is significant; we included a range of BF thresholds from 1/10 to 200 |
| (6) Bayesian meta-analysis | original study and fixed-effect Bayesian meta-analysis have a positive effect size, in addition to the JZS BF of the meta-analysis being larger than a BF threshold; we included a range of BF thresholds from 1/10 to 200 |
| (7) sceptical *p*-value | original study has a positive effect size, in addition to sceptical *p*-value being significant; we included a range of alpha levels from 0.0001 to 0.5 |

effect when there is *not*. That is, for each metric, criteria have been formulated telling us whether we should conclude that the finding has been successfully replicated or not. Finally, contrary to Schauer & Hedges [22], we explicitly include publication bias into our simulations and examine how it affects the performance of the different metrics.

In this paper, we will present a simulation study to formally investigate the performance of different metrics to answer the question: 'How do different metrics of replication success compare in terms of detecting real effects (true positive rate) and detecting spurious effects (false positive rate) across various levels of publication bias?'. To answer this research question, we selected the following replication success metrics based on our replication literature review: the traditional significance test [30], the Small Telescopes [31], the (Bayesian) meta-analysis [30,32], the default BF [33,34], the replication BF [21] and the sceptical *p*-value [35]. These quantitative measures of replication success focus on hypothesis testing and make claims about the existence of a true underlying population effect.

In what follows, we will first describe each of these metrics in more detail. Then, we will describe the set-up of our simulation study. In the results section, we will compare the performance of each of these metrics in terms of true and false positive rates. In the discussion, we will discuss implications for replication researchers.

# 2. Method

## 2.1. Replication success metrics

In table 1, the metrics of replication success and their associated criteria as operationalized in this study are summarized. Below, we will describe them in more detail. We mostly follow replication studies that use these measures in order to quantify replication success. Specifically, we follow the procedures applied by large replication projects such as Camerer *et al.* [12,13], Cova *et al.* [15] and OSC [18]. In these studies, replication success is usually operationalized as a positive result in the replication attempt following a

positive result in the original study (original null results are replicated less often, since there are far fewer original null results in the scientific literature). In a sense, the original study result is the golden standard against which to anchor the replication. In a simulation study, one has the advantage of knowing the true state of the world. Rather than using the original sample result as a proxy for the true population effect, one can evaluate the replication attempt against the 'known truth' directly. For this reason, in addition to assessing replication success, we go one step further, namely identifying true and false positives. In the context of our study, a true positive refers to obtaining a positive replication result when the underlying true effect is non-zero; a false positive refers to obtaining a positive replication result when the underlying true effect is practically zero.

### 2.1.1. Significance

Traditionally, the evaluation of a replication attempt has been based on the classical null hypothesis significance test (NHST). Specifically, a replication attempt is considered to be successful when both the original and replication study report a significant $p$-value, in addition to both studies having an effect in the same direction (e.g. [30,36]). This measure is known as 'vote counting' [30]. In this simulation study, we operationalize replication success for this method as a significant result in the replication study alone in order to be able to apply this method to original studies without a significant result as well.

### 2.1.2. Small Telescopes

An approach that combines hypothesis testing with effect size estimation is the Small Telescopes, proposed by Simonsohn [31]. This method consists of two steps. In the first step, a small effect size is defined that gives a low power to the original study. In our simulation study, we follow Simonsohn [31] who chose a power level of 33% for this approach. In the second step, the effect size of the replication study is estimated, followed by testing whether the estimated replication effect size is significantly smaller than the small effect size defined in the first step.

### 2.1.3. Classical meta-analysis

A different way to approach the question of how to quantify replication success is to conduct a meta-analysis by combining the original study and the replication attempt(s) to estimate their overall effect size (e.g. [30]). Although meta-analysis has often been used in the replication literature (e.g. [12,13,18]), a frequently mentioned threat to the validity of this method is publication bias (e.g. [37]), since it weighs in the outcomes from the original, and possibly biased, study.

### 2.1.4. Bayes factor

The Bayesian counterpart of the NHST is the default Jeffreys–Zellner–Siow (or JZS) BF that yields support in favour of or against the null hypothesis relative to the alternative hypothesis [33,34]. Of particular relevance for replication literature is that this test quantifies evidence for the effect being absent relative to the effect being present in the replication study [21].

### 2.1.5. Replication Bayes factor

Verhagen & Wagenmakers [21] have argued that the default JZS BF hypothesis test is not relating the data from the original study to the data from the replication study. As an alternative, Verhagen & Wagenmakers [21] have proposed the replication BF. This BF quantifies evidence for the effect in the replication attempt being absent relative to the effect being similar to the effect that was found in the original experiment. It is important to note that this metric can only be used if the original result is significant [21].

### 2.1.6. Bayesian meta-analysis

The Bayesian equivalent of the classical meta-analysis (e.g. [32]) quantifies evidence for the effect being absent relative to the effect being present across both studies [21].

### 2.1.7. Sceptical *p*-value

The rationale behind the sceptical *p*-value [35] is to challenge the original experiment by questioning how sceptical one needs to be *a priori* to not be convinced by its result that the effect exists. Specifically, this method constructs a prior that is sufficiently sceptical such that, combined with the original data, a posterior is constructed for which the associated credible interval just overlaps with zero. Subsequently, the extent to which the replication data contradicts this sceptical prior is quantified with the sceptical *p*-value [35].

## 2.2. Underlying true population effects

To compare the different quantitative measures of replication success in terms of detecting real (i.e. obtaining replication success when the underlying true effect is non-zero) and spurious effects (i.e. obtaining replication success when the underlying true effect is virtually zero), 20 000 standardized population effect sizes were randomly drawn from a zero (spurious) and a non-zero (real) population effect size distribution. Both distributions are normal with mean $\mu$, and standard deviation $\tau$. For the spurious effects, $\mu = 0$ and $\tau = 0.02$; for the real effects, $\mu = 0.5$ and $\tau = 0.15$. The rationale behind the decision of randomly drawing 20 000 standardized population effect sizes from each distribution, rather than using fixed underlying true effect sizes, is to obtain variation across the true population effect sizes. Additionally, for the spurious effects, a defensible choice would have been to draw samples from populations with a mean exactly equal to zero. We opted, instead, for effect sizes situated very close to zero to allow for the fact that in practice, negligible effects may well not be exactly zero.

## 2.3. Original studies

For each of the 20 000 population effect sizes in both the spurious and real effect conditions, sample data were generated of sizes $n = 25, 50, 75$ or 100. Thus, per condition, per sample size, 5000 datasets were sampled. The data were randomly generated from a normal distribution centred on truth, with a constant standard deviation of 1. For each dataset, a one-sample *t*-test was conducted, where the null hypothesis was $H_0: \mu = 0$ and the alternative hypothesis was $H_a: \mu > 0$ (i.e. one-tailed).

## 2.4. Publication bias

We applied a set of five models for the publication process: 0%, 25%, 50%, 75% or 100% publication bias. The 100% publication bias model reflects the situation in which only significant original studies (with a nominal alpha level of 0.05) are published. We realize that this model is unrealistic [38], given occasional non-significant findings in scientific literature. In the 0% model, all original studies are published regardless of significance. Although scientifically optimal, this model is not realistic either [38]. In the remaining three models, the nominally significant initial studies are always published, while the non-significant originals are published based on a constant probability. For example, in the 75% publication bias model, all significant original experiments are published, while only 25% of the non-significant studies are published. Since, of course, in practice not all significant results are published either, the publication bias percentages could be read as the percentage of non-significant findings that are published relative to the percentage of significant findings that are published. The publication bias process is clarified in figure 1.

## 2.5. Replication studies

For each original published study, a single close replication study was conducted in a similar way to the original attempt. The sample size of the replication studies was twice as large as the sample size of the initial studies, meaning that the sample size of the replication studies was $n = 50, 100, 150$ or 200. This was done to be consistent with the fact that replication attempts tend to have a larger sample size than their original counterparts (e.g. [12,13,15,18]).

## 2.6. True and false positives

For each condition, for a range of threshold values, we computed the *proportion of true positives* as the proportion of positive findings out of the total set of findings from populations with a *non-zero* mean

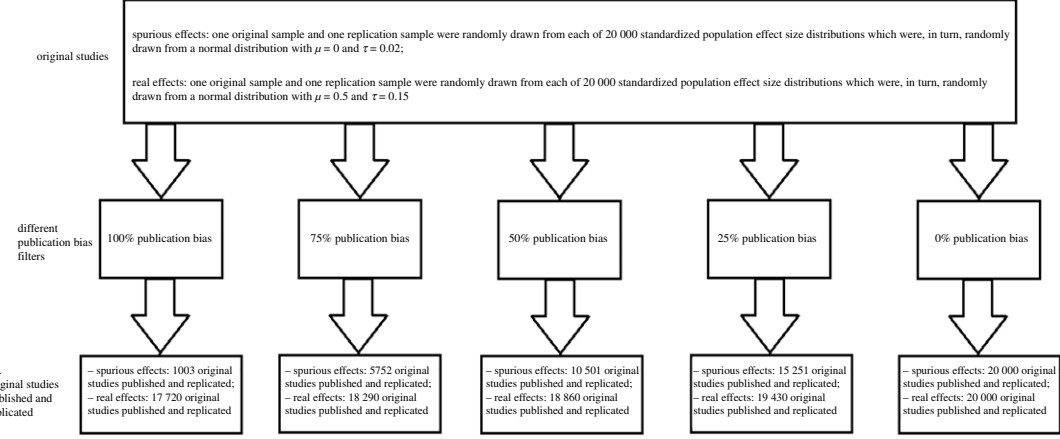

**Figure 1.** Operationalization of publication bias. For each of the different values of publication bias (i.e. 100, 75, 50, 25 and 0%), all original studies were selected for which the resulting $p$-value was less than 0.05 (i.e. published and replicated). As for the original studies with a $p$-value $\geq$ 0.05, we included a fraction of them depending on the level of simulated publication bias as follows: 100% publication bias: 0% of the non-significant original studies were selected; 75% publication bias: 25% of the non-significant original studies were selected; 50% publication bias: 50% of the non-significant original studies were selected; 25% publication bias: 75% of the non-significant original studies were selected; 0% publication bias: 100% of the non-significant original studies were selected.

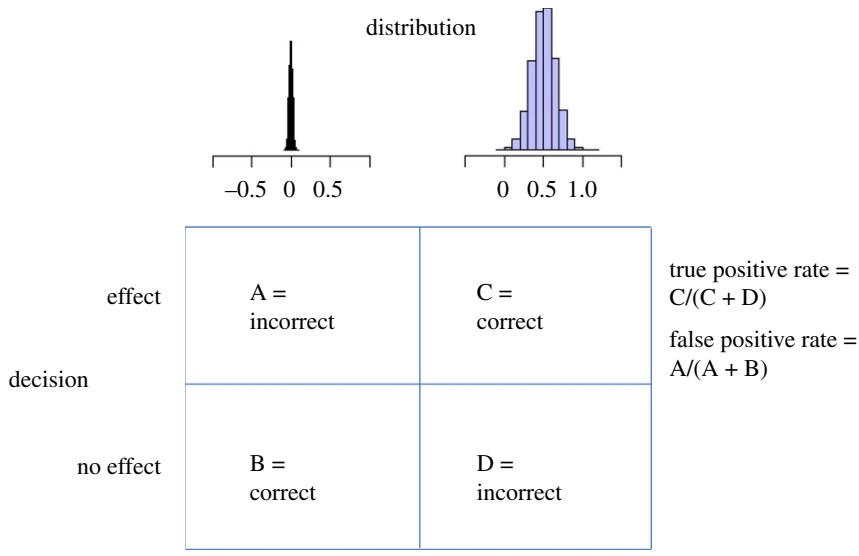

**Figure 2.** False positive rate = the number of positive findings (cell A) out of the total set of findings from populations with a ($\mu = 0$, $\tau = 0.02$) distribution (cell A + B). True positive rate = the number of positive findings (cell C) out of the total set of findings from populations with a ($\mu = 0.50$, $\tau = 0.15$) distribution (cell C + D).

distribution. Likewise, we computed the *proportion of false positives* as the proportion of positive findings out of the total set of findings from populations with a *zero* mean distribution (figure 2). Without publication bias, both proportions are based on 20 000 findings. However, in the presence of publication bias, we have obtained fewer findings, and especially the number of findings from zero mean distributions may decrease considerably (to approximately 1000 in case of 100% publication bias). We computed these proportions for each of the seven metrics. Note, however, that for the replication BF, it is required to have a significant result in the original study, and this is only guaranteed in the condition of 100% publication bias. Therefore, for the other publication bias conditions, no results could be computed for this metric. All parameter values are given in table 2, and examples of a true and a false positive finding for each metric can be found in table 3.

**Table 2.** Summary of parameter values included in the simulation.

| parameter | definition | simulated values |
|---|---|---|
| Iter | number of iterations per simulation cell | 5000 |
| n | sample size original studies | 25, 50, 75, 100 |
| n.rep | sample size replication studies | 50, 100, 150, 200 |
| truth | true population effect size distribution | $\sim N(\text{mean} = 0, \text{s.d.} = 0.02)$ |
|  |  | $\sim N(\text{mean} = 0.5, \text{s.d.} = 0.15)$ |
| publication bias | percentage of non-significant findings that are published relative to the percentage of significant findings that are published | 100, 75, 50, 25, 0 |

# 3. Results

The proportion of true and false positive rates for the different replication success metrics across different publication bias levels are presented in table 4. For both the frequentist and Bayesian approaches, we used liberal thresholds, which have been reported in the replication literature as well. This means that for the frequentist approaches, we used an alpha level of 0.05, and for the Bayesian approaches, we used a BF threshold of 1. We started with a single threshold for both the frequentist and Bayesian metrics to be able to make rough comparisons of the true and false positive rates across the replication success metrics.

Several general observations can be made when comparing the replication success metrics with one another across the five different publication bias levels. These should, however, be made taking into account uncertainty margins around the observations. Rough estimates of 95% uncertainty margins[1] for *proportions of true positives* are $\pm 0.0030$ (i.e. $1.96 \times \sqrt{[0.05 \times 0.95/20\,000]}$), while those for the *proportions of false positives* range from $\pm 0.0030$ (i.e. $1.96 \times \sqrt{[0.05 \times 0.95/20\,000]}$) to $\pm 0.0135$ (i.e. $1.96 \times \sqrt{[0.05 \times 0.95/1000]}$) for no versus maximal publication bias, respectively. These margins are for proportions, hence for percentages we have $\pm 0.3\%$ and $\pm 1.4\%$, respectively. Findings in table 4 seem to show a decrease in both the true and false positive rates when the publication bias level decreases. For significance, Small Telescopes, BF, (Bayesian) meta-analysis and sceptical *p*-value, the false positive rate shows the largest decrease when moving from 100% to 75% publication bias, followed by a smaller and more gradual decrease for the remaining decreasing publication bias levels. On the other hand, the true positive rate does not show such a large drop when moving from the 100% to the 75% publication bias scenario.

The significance metric and the JZS BF seem to have similar results. The false positive rate is approximately 5 to 6% when publication bias is 100%, while this rate is about 2 to 3% for the remaining publication bias levels. The true positive rate for these two metrics varies between approximately 95% for the lower publication bias levels and 98% for the higher publication bias levels.

The Small Telescopes has much higher false positive rates compared with the remaining metrics, which varies from 24% when publication bias is 0% to 51.7% when publication bias is 100%. Although the true positive rate shows a little decrease with decreasing levels of publication bias, this rate is almost 100% across all publication bias scenarios.

The classical meta-analysis has a high false positive rate (40.5%) when publication bias is 100%, while this rate is about 6 to 11% for the remaining publication bias levels. The true positive rate for this metric varies between approximately 98% for the lower publication bias levels and 100% for the higher publication bias levels.

Looking at the Bayesian meta-analysis, the false positive rate is 29.4% when publication bias is 100%, while this rate is about 4 to 7% for the remaining publication bias levels. The true positive rate varies between approximately 97% for the lower publication bias levels and 100% for the higher publication bias levels.

---

[1]The provided 95% uncertainty margins are based on the assumption of true proportions being equal to 0.95 (or 0.05); for proportions of 0.99 (or 0.01), the uncertainty margin is less than half. The fact that results are mostly qualitatively similar across publication bias simulation sets suggests the reported uncertainty margin is likely to be an overestimate.

**Table 3.** Examples of a true and false positive finding for each metric.

| metric | false positive | true positive |
|---|---|---|
| (1) significance | an original study and its replication attempt both have a positive observed effect size; the replication attempt has a *p*-value below 0.05 for a known population effect size of (approximately) 0 | an original study and its replication attempt both have a positive observed effect size; the replication attempt has a *p*-value below 0.05 for a known population effect size greater than 0 |
| (2) Small Telescopes | an original observed effect size is positive; the observed replication effect size is not significantly smaller than the small effect size, which is computed based on the observed original effect size for a known population effect size of (approximately) 0 | an original observed effect size is positive; the observed replication effect size is not significantly smaller than the small effect size, which is computed based on the observed original effect size for a known population effect size greater than 0 |
| (3) classical meta-analysis | an observed original effect size and the fixed-effect meta-analytic effect size are both positive; the meta-analysis has a *p*-value below 0.05 for a known population effect size of (approximately) 0 | an observed original effect size and the fixed-effect meta-analytic effect size are both positive; the meta-analysis has a *p*-value below 0.05 for a known population effect size greater than 0 |
| (4) BF | an original study and its replication attempt both have a positive observed effect size; the BF of the replication study is larger than 1 for a known population effect size of (approximately) 0 | an original study and its replication attempt both have a positive observed effect size; the BF of the replication study is larger than 1 for a known population effect size greater than 0 |
| (5) replication BF | an original study has a *p*-value below 0.05; the replication BF is larger than 1 for a known population effect size of (approximately) 0 | an original study has a *p*-value below 0.05; the replication BF is larger than 1 for a known population effect size greater than 0 |
| (6) Bayesian meta-analysis | an observed original effect size and the Bayesian meta-analytic effect size are both positive; the meta-analysis has a BF larger than 1 for a known population effect size of (approximately) 0 | an observed original effect size and the Bayesian meta-analytic effect size are both positive; the meta-analysis has a BF larger than 1 for a known population effect size greater than 0 |
| (7) sceptical *p*-value | an observed original effect size is positive; the sceptical *p*-value for an original study and its replication attempt is smaller than 0.025 for a known population effect size of (approximately) 0 | an observed original effect size is positive; the sceptical *p*-value for an original study and its replication attempt is smaller than 0.025 for a known population effect size greater than 0 |

As explained above, for the replication BF, results were only computed for the 100% publication bias level. Looking at the replication BF, the false positive rate is somewhat high (9.8%). It is clearly lower than that for the Small Telescopes and meta-analysis metrics, but clearly higher than that for the significance, JZS BF and sceptical *p*-value metrics. The true positive rate is quite high (97.6%) and comparable to that for the significance, JZS BF and sceptical *p*-value metrics, but lower than for the Small Telescopes and meta-analysis metrics.

For the sceptical *p*-value, the false positive rate is 4.7% when publication bias is 100%, while this rate is about 0.3 to 0.8% for the remaining publication bias levels. The true positive rate varies between approximately 89% for the lower publication bias levels and 98% for the higher publication bias levels.

**Table 4.** Percentages replication success across different publication bias levels. *Note:* percentages replication success for the columns where truth is 0 represent percentages false positives; percentages replication success for the columns where truth is 0.5 represent percentages true positives.

| replication success criteria | publication bias 100% | | publication bias 75% | | publication bias 50% | | publication bias 25% | | publication bias 0% | |
| --- | --- | --- | --- | --- | --- | --- | --- | --- | --- | --- |
| | truth 0 (%) | truth 0.5 (%) | truth 0 (%) | truth 0.5 (%) | truth 0 (%) | truth 0.5 (%) | truth 0 (%) | truth 0.5 (%) | truth 0 (%) | truth 0.5 (%) |
| significance ($p < 0.05$) | 6.2 | 97.9 | 3.1 | 97.2 | 3.1 | 96.6 | 2.8 | 96 | 2.8 | 95.4 |
| Small Telescopes ($p > 0.05$) | 51.7 | 99.9 | 28.1 | 99.5 | 25.1 | 99.3 | 24.7 | 99.1 | 24 | 98.8 |
| classical meta-analysis ($p < 0.05$) | 40.5 | 99.8 | 10.5 | 99.2 | 7.7 | 98.6 | 6.4 | 98.2 | 5.9 | 97.6 |
| Bayes factor (BF > 1) | 5.2 | 97.7 | 2.3 | 97 | 2.3 | 96.4 | 2 | 95.8 | 2 | 95.2 |
| replication Bayes factor (BF > 1) | 9.8 | 97.6 | — | — | — | — | — | — | — | — |
| Bayesian meta-analysis (BF > 1) | 29.4 | 99.7 | 7 | 99 | 5 | 98.4 | 4 | 97.9 | 3.7 | 97.3 |
| sceptical $p$-value ($ps < 0.025$) | 4.7 | 97.9 | 0.8 | 95.5 | 0.5 | 93.3 | 0.3 | 91 | 0.3 | 89.1 |

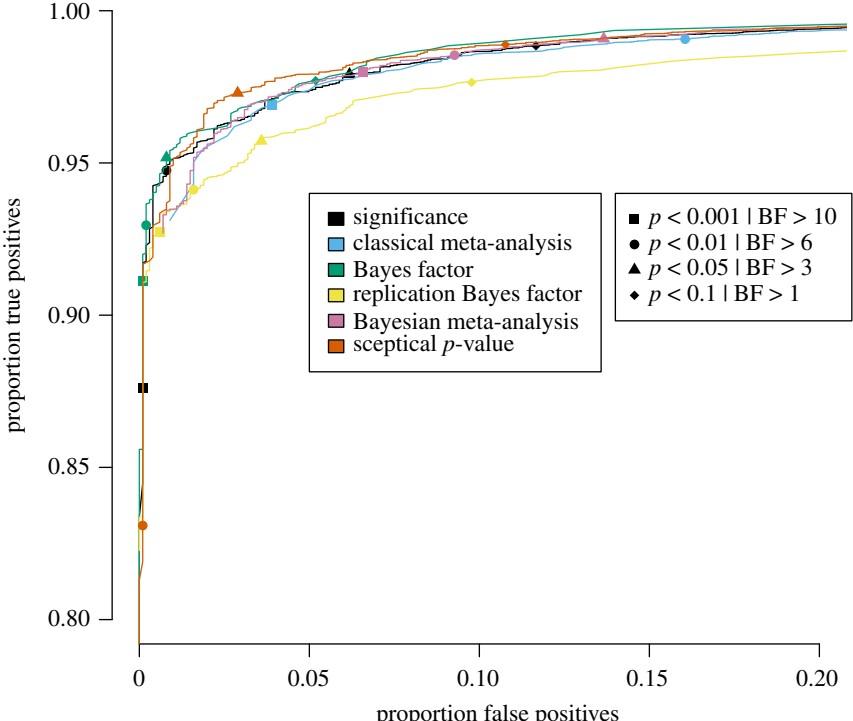

**Figure 3.** Proportion of true positives plotted against proportion of false positives for publication bias level of 100%, meaning that 0% of the non-significant original studies were published and replicated. Different replication success metrics are indicated by different colours; different symbols indicate different thresholds of the replication success metrics. Significance thresholds range from 0.0001 (bottom-left) to 0.5 (top-right), BF thresholds range from 200 (bottom-left) to 1/10 (top-right).

Overall, these findings show that the Small Telescopes[2] did qualitatively worse in terms of a high rate of false positives compared with the remaining metrics. However, it might be difficult to compare the significance metric, the (replication) BF, the (Bayesian) meta-analysis and the sceptical $p$-value with one another based on table 4. Therefore, we decided to more thoroughly examine the differences in true and false positive rates of these six methods to be able to make better comparisons by including a wide range of decision criteria for each method.

Specifically, for each of the five publication bias level scenarios, the proportion of true positives is plotted against the proportion of false positives for a wide range of threshold values in figures 3–7 for the traditional significance test; the (Bayesian) meta-analysis; the JZS BF; the replication BF and the sceptical $p$-value. In figures 3–7, the y-axis represents the proportion of true positives and the x-axis represents the proportion of false positives. Different figures indicate different publication bias levels, and different colours represent different quantitative measures of replication success. The ideal scenario of detecting 100% true positives and 0% false positives is represented by the top left of each figure.

For the frequentist methods and the sceptical $p$-value, the alpha levels varied from 0.0001 to 0.5 in increments of 0.0001 (i.e. 5000 thresholds). For the Bayesian methods, the BF thresholds varied from 1/10 to 200 in increments of 1/25 (i.e. 4998 thresholds). We do realize that the majority of thresholds we included in our study are not used in practice, and in the case of the Bayesian methods, even provide evidence for the null hypothesis. However, the aim of the present examination is not to compare individual thresholds with one another, but to compare the different quantitative measures of replication success to one another across the board. For instance, if the curve for the JZS BF is entirely above and to the left of the curve for the replication BF, that indicates that for any

---

[2]We have repeated the simulation for replications with a sample size 2.5 times the original sample size as per the recommendations of Simonsohn [31]. Although the percentage of true positives remained practically the same, the percentage of false positives did decrease slightly. However, these results were still qualitatively worse compared with the results of the remaining metrics. For publication bias ranging from 100% to 0% in decreasing steps of 25%, the percentages of false positives were 43.8%, 23.7%, 21.4%, 21.1% and 20.6%, respectively.

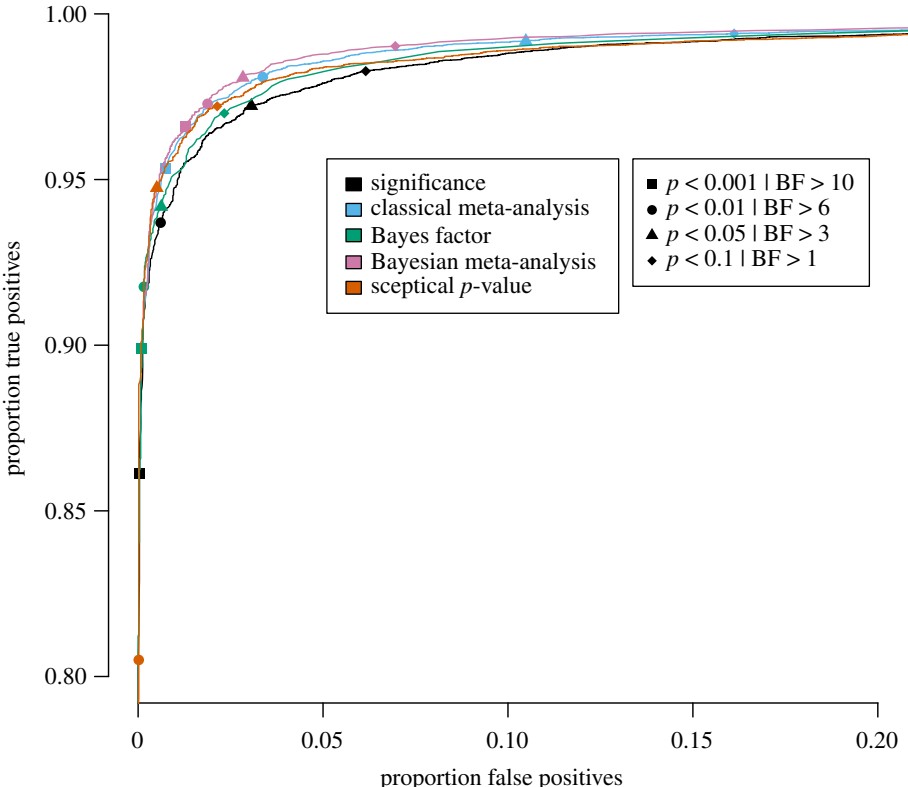

**Figure 4.** Proportion of true positives plotted against proportion of false positives for publication bias level of 75%, meaning that 25% of the non-significant original studies were published and replicated. Different replication success metrics are indicated by different colours; different symbols indicate different thresholds of the replication success metrics. Significance thresholds range from 0.0001 (bottom-left) to 0.5 (top-right), BF thresholds range from 200 (bottom-left) to 1/10 (top-right).

threshold for the replication BF, there exists a threshold for the JZS BF that has a higher true positive rate *and* a lower false positive rate and is therefore strictly better in terms of quantifying replication success. To provide the reader with a handhold, we marked commonly used thresholds (frequentist and sceptical *p*-value thresholds: alpha = 0.001, 0.01, 0.05 and 0.1; Bayesian thresholds: BF = 1, 3, 6 and 10) in the figures with specific symbols.

Several general observations can be made when comparing the five different figures. These should again be made taking into account our rough estimates of uncertainty margins around the observations (see above: ±0.3% for the true positive rates, and between ±0.3% and ±1.4% for the false positive rates, depending on the percentage of publication bias). The findings show the extent to which working with more conservative thresholds for statistical evidence reduces the false positive rates at the expense of reducing the true positive rates.

When looking at the top left in figure 3, which represents 100% publication bias, it can be observed that the sceptical *p*-value seems to outperform all other metrics in the panel; however, the curves for the sceptical *p*-value and the JZS BF appear to cross each other (approximately, x-axis is 0.025 and y-axis is 0.96): the sceptical *p*-value seems to perform better than the BF for the more liberal thresholds in the top left, whereas the BF seems to perform slightly better for the more conservative thresholds. Furthermore, the JZS BF seems to outperform the significance metric, the (Bayesian) meta-analysis and the replication BF, though the differences between the JZS BF and the significance metric are quite small. Although the classical and Bayesian meta-analyses seem to have a similar performance, the Bayesian meta-analysis lies just slightly more to the top and more to the left of the frequentist alternative. Both meta-analyses seem to perform slightly worse than the significance method for the more conservative thresholds. The replication BF clearly performs worse with respect to true and false positive rates than the sceptical *p*-value, the JZS BF and the significance metric, with differences clearly exceeding the uncertainty margins. Although the replication BF performs worse than both meta-analyses, the curve for the replication BF on the one hand and the curves for the meta-analytical techniques on the other hand appear to cross each other (approximately, x-axis is 0.02 and y-axis is 0.93). Specifically, the replication BF

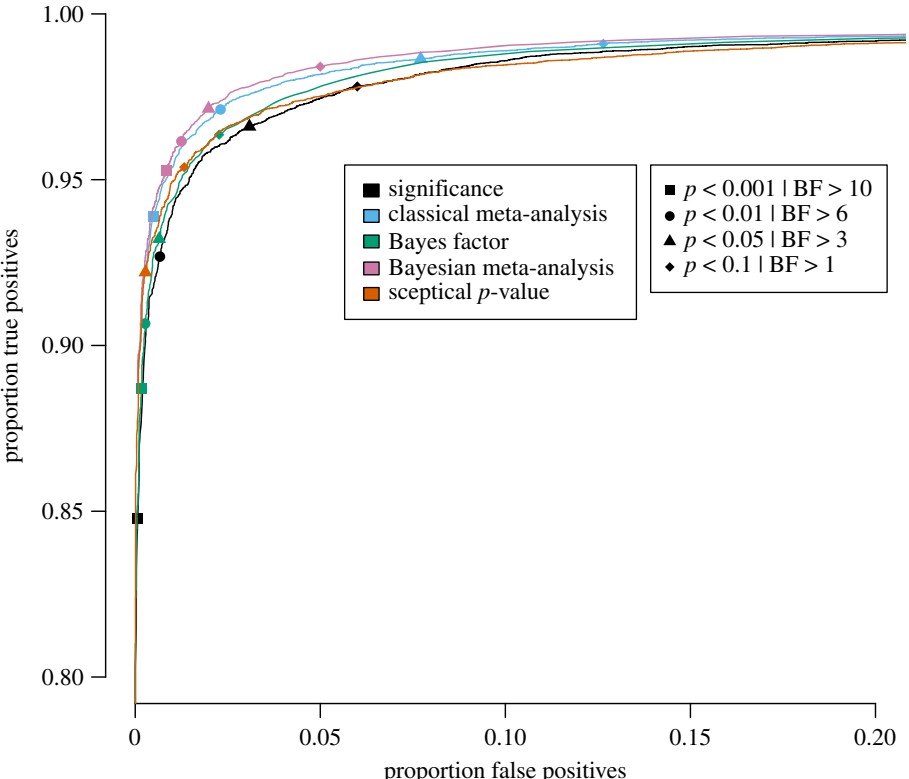

**Figure 5.** Proportion of true positives plotted against proportion of false positives for publication bias level of 50%, meaning that 50% of the non-significant original studies were published and replicated. Different replication success metrics are indicated by different colours; different symbols indicate different thresholds of the replication success metrics. Significance thresholds range from 0.0001 (bottom-left) to 0.5 (top-right), BF thresholds range from 200 (bottom-left) to 1/10 (top-right).

performs better than the classical meta-analysis, but similar to the Bayesian meta-analysis for the very conservative thresholds.

Looking at figure 4, where publication bias is 75%, some changes can be observed compared with the previous findings. The Bayesian meta-analysis seems to have the best performance in terms of true and false positive rates, followed by the classical meta-analysis, the sceptical $p$-value, the JZS BF and the significance metric. Just like for the 100% publication bias scenario, these differences are fairly small. Note that the replication BF is designed for scenarios where the original result is significant. Therefore, findings for the replication BF are only presented for the scenario of 100% publication bias.

When publication bias is 50%, we find a similar pattern to the 75% publication bias scenario (figure 5), except for the sceptical $p$-value. This means that the Bayesian meta-analysis has the best performance, followed by the classical meta-analysis, the JZS BF, the sceptical $p$-value (the latter two seem to have a very similar performance in the top left of the figure) and the significance metric.

Quite consistent with the previous two publication bias levels, figure 6 shows that when publication bias is 25%, the Bayesian meta-analysis is again the best-performing metric, followed by the classical meta-analysis, the default BF, the sceptical $p$-value and the significance metric (the latter two have a very similar performance).

Finally, figure 7 displays the results for an idealistic scenario in which no publication bias exists. The pattern of results mirrors those of publication bias levels 25%, 50% and 75%, except for sceptical $p$-value. Bayesian meta-analysis has the best performance, followed by the classical meta-analysis, the default BF, the significance metric and the sceptical $p$-value, with the differences between the methods again being relatively small.

To provide one, fairly arbitrary, example of a comparison between two methods for specific thresholds: inspection of each figure shows that the green triangle (corresponding to a JZS BF with a threshold of 3) is right above the black circle (corresponding to the significance metric with alpha of 0.01) in each of the five figures. This indicates that regardless of the level of publication bias, the JZS BF with a threshold of 3 obtains a higher rate of true positives (approximately 0.3–0.5%) *and* an equal rate of false positives compared with the significance metric at threshold 0.01.

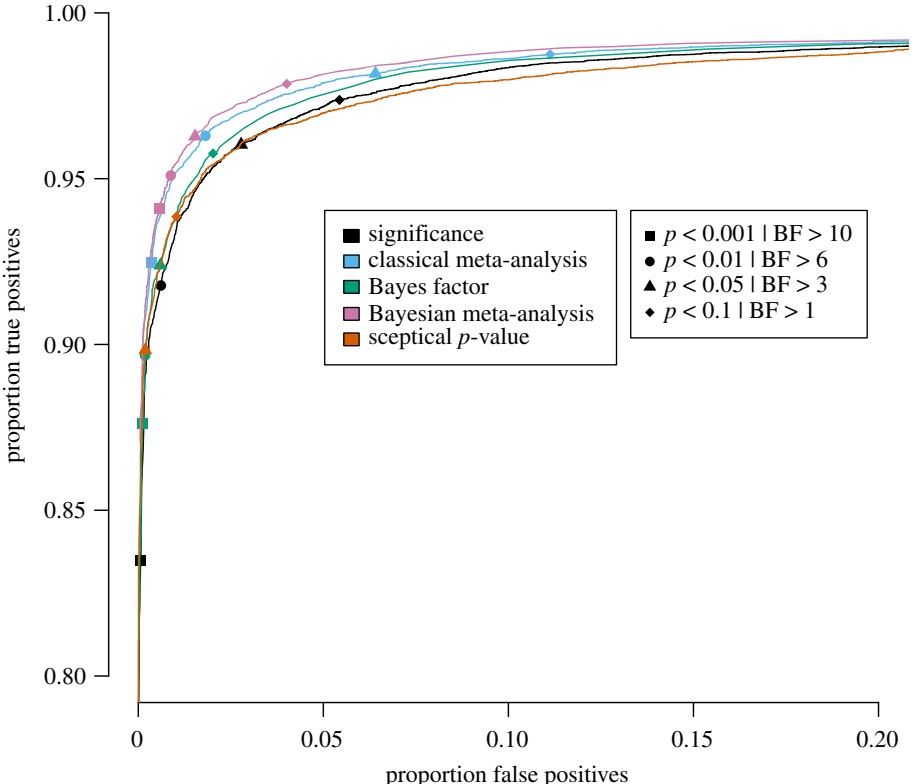

**Figure 6.** Proportion of true positives plotted against proportion of false positives for publication bias level of 25%, meaning that 75% of the non-significant original studies were published and replicated. Different replication success metrics are indicated by different colours; different symbols indicate different thresholds of the replication success metrics. Significance thresholds range from 0.0001 (bottom-left) to 0.5 (top-right), BF thresholds range from 200 (bottom-left) to 1/10 (top-right).

In sum, Bayesian metrics seem to slightly outperform frequentist metrics across the board. Generally, meta-analytic approaches seem to slightly outperform metrics that evaluate single studies, except in the scenario of 100% publication bias, where this pattern reverses.

## 4. Discussion

In the present study, we simulated original and replication studies to evaluate and compare a number of quantitative measures of replication success in terms of detecting real and spurious effects across different publication bias levels. We included the following replication success metrics that make claims about the existence of a true underlying population effect: the traditional significance test metric, the Small Telescopes, the (Bayesian) meta-analysis, the default BF, the replication BF and the sceptical *p*-value.

The take-home message when surveying the results of all simulations together is that the sceptical *p*-value and the JZS BF seem to have the best performance with respect to true and false positives when one assumes that only significant original studies achieve publication. In the scenarios where we assumed that at least 25% of the non-significant original studies achieve publication, the Bayesian meta-analysis seems to outperform the remaining replication success metrics. However, it is important to note that the differences in true and false positive rates between the significance metric, the JZS BF, the (Bayesian) meta-analysis and the sceptical *p*-value were quite small. The replication BF performed slightly worse compared with the significance metric, the JZS BF, the (Bayesian) meta-analysis and the sceptical *p*-value. The Small Telescopes did worse compared with the other metrics; it seemed to be too liberal.

Replication success is an interesting concept; when conducting the present study we also learned that replication success is a difficult concept. We used replication success in this study in its binary form (i.e. success/failure). However, one could argue that the extent to which the findings of two studies are consistent with each other is a matter of degree, and binary success/failure inferences might be overly

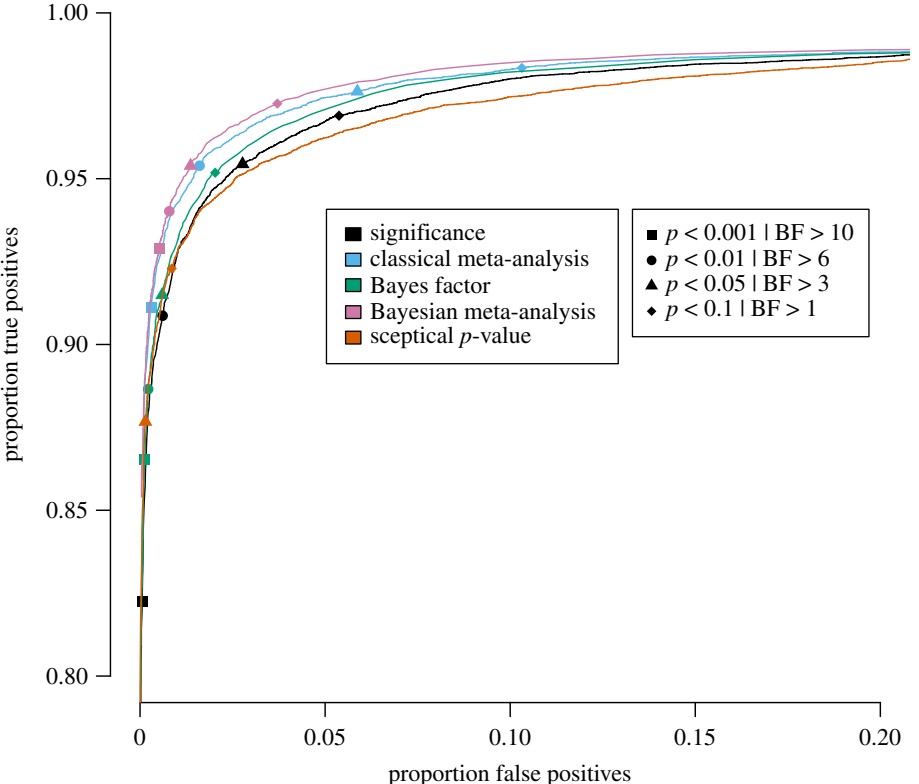

**Figure 7.** Proportion of true positives plotted against proportion of false positives for publication bias level of 0%, meaning that 100% of the non-significant original studies were published and replicated. Different replication success metrics are indicated by different colours; different symbols indicate different thresholds of the replication success metrics. Significance thresholds range from 0.0001 (bottom-left) to 0.5 (top-right), BF thresholds range from 200 (bottom-left) to 1/10 (top-right).

simplistic. Additionally, a success/failure inference based only on consideration of the quantitative results of a study ignores other important factors such as study design. In our study, the original and the replication studies were exactly the same; the only differences between the two were the samples (i.e. for the replication study, a new sample was randomly drawn from the population) and the sample sizes (i.e. replication sample size was twice as large as the original sample size). However, in daily science, a replication classified as 'failed' might not necessarily mean that the tested theory is wrong. In fact, it has been argued that the failure to replicate results might reflect the presence of hidden moderators, which refers to contextual differences between the original study and the replication attempt (e.g. [39]). Finally, one could argue whether it is interesting to compare the findings of two studies rather than all relevant studies exploring the research question of interest (e.g. via meta-analysis).[3]

Several caveats need to be discussed. Not all available quantitative measures of replication success were included; some of them did not fit in the design of the present study. For example, Anderson & Maxwell [36] suggest using theory and past research to construct a confidence interval. Since we conducted a simulation study, only metrics were included for which such theoretical considerations are not required, because the simulated 'phenomena' were basically numbers representing fictitious studies, and hence theory-free. Furthermore, the metrics included in our study focus on hypothesis testing; two additional metrics that focus on estimation are the 'coverage' [18,30] and the prediction interval [40]. In contrast with the testing metrics, these two methods do not make claims about the existence of a true underlying population effect; they simply determine the comparability and the consistency between the findings of the original and the replication study [30,40]. Although the coverage and the prediction interval have been widely used in large replication projects [12,13,15,18], we did not include them in our study, because the notion of true and false positives is not appropriate for either of these methods.

---

[3]We thank an anonymous reviewer for pointing out the information provided in this paragraph.

Furthermore, it should be noted that some of the methodological decisions were made for the sake of the simplicity of the present study. The first one is that we included only two distributions for the underlying true population effect sizes, namely one distribution where the mean was 0.5 (for the real effects), and one distribution where the mean was 0 (for the spurious effects). Even though we used distributions in our simulations (meaning that there were various true underlying effect sizes), rather than fixed underlying true effect sizes, and incorporated what we believe to be a realistic range of effect sizes, we cannot exclude the possibility that different choices for our effect size distributions would have led to different results. Additionally, we would like to emphasize that all procedures in this study are being evaluated on a criterion that we would like them to perform well on (separating meaningful from spurious true effects), but not on a criterion they was necessarily designed to perform well on.

Another factor that might affect our results is the operationalization we used for publication bias. We included five constant probabilities of publishing non-significant results, regardless of the size of the $p$-value. It could be argued that the mechanism behind the publication process is more complex than this. Several other, more gradual, biasing mechanisms are discussed by Guan & Vanderkerckhove [38].

Furthermore, in the present study, we simulated that all replication studies were published, so there was no publication bias for the replication attempts. This may sound like an unrealistic scenario, since journals mostly seem to accept studies that are novel, meaning they prefer original studies over replications (e.g. [25]). However, on the other hand, in large replication projects (e.g. [12,13,15,18]), no publication bias exists for the replication studies, since all findings for all replication attempts are reported.

Finally, it is important to note that our results might depend on how we defined the criteria for each metric. We do not think that our results would be qualitatively different if we would have defined the replication success criteria differently, but we cannot rule this out.

Allowing for these caveats, this study is one of the first attempts to compare different quantitative measures of replication success in terms of detecting real and spurious effects. Even though the differences between the majority of the methods were small, the JZS BF and the Bayesian meta-analysis had higher true positives and/or lower false positives than their frequentist counterparts, irrespective of the publication bias level. The sceptical $p$-value performed particularly well under scenarios of high publication bias.

Data accessibility. The dataset supporting this manuscript can be obtained by running the available R codes (see 'Code availability' below). However, we would like to provide you with the dataset simulated by us as well, which can be found in the R file '2021-02-15 Results.RData' uploaded as part of the supplementary material.

Code Availability. For the data simulation, the R file '2021-02-15 Data Simulation.R' can be used; for figures 3–7, the R file '2021-02-15 ROC plots.R' can be used. Both R codes were written in R v. 3.6.1, and are uploaded as part of the electronic supplementary material. Data and R codes are available within the OSF repository: osf.io/p9wcn/ [41].

Authors' contributions. J.M., R.H., H.K. and D.v.R. meet the following authorship conditions: substantial contributions to conception and design, or acquisition of data, or analysis and interpretation of data; drafting the article or revising it critically for important intellectual content; final approval of the version to be published; and agreement to be accountable for all aspects of the work in ensuring that questions related to the accuracy or integrity of any part of the work are appropriately investigated and resolved. J.M. participated in data/statistical analysis, participated in the design of the study, drafted the manuscript and critically revised the manuscript; R.H. participated in the design of the study and critically revised the manuscript; H.K. participated in the design of the study and critically revised the manuscript; D.v.R. participated in data/statistical analysis, participated in the design of the study and critically revised the manuscript. All authors gave final approval for publication and agree to be held accountable for the work performed therein.

Competing interests. There are no conflicting or competing interests related to the present manuscript.

Funding. This work was supported by the University of Groningen.

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
