## [Peer Review File · Royal Society Open Science]

Review History

RSOS-201697.R0 (Original submission)

Review form: Reviewer 1 (Peder Isager)

Is the manuscript scientifically sound in its present form?

No

Are the interpretations and conclusions justified by the results?

Yes

Is the language acceptable?

No

Do you have any ethical concerns with this paper?

No

Have you any concerns about statistical analyses in this paper?

Yes

Recommendation?

Major revision is needed (please make suggestions in comments)

Comments to the Author(s)

See reviewer notes (Appendix A).

Review form: Reviewer 2

Is the manuscript scientifically sound in its present form?

Yes

Are the interpretations and conclusions justified by the results?

Yes

Is the language acceptable?

Yes

Do you have any ethical concerns with this paper?

No

Have you any concerns about statistical analyses in this paper?

No

Recommendation?

Major revision is needed (please make suggestions in comments)

Comments to the Author(s)

Summary

This article reports a simulation study designed to compare different measures of 'replication success'. The manuscript is written clearly and concisely. The study appears technically sound; however, I would like to see a more detailed justification for various conceptual tenants of the project (see 'major comments' below). The study has commendable adherence to transparency - all data and analysis code have been made available. The study's conclusions appear well calibrated to the evidence and relevant limitations are appropriately acknowledged and discussed.

Major comments

- I felt the introduction could do more to justify why replication success is an interesting concept to explore. One could argue, for example, that the extent to which the findings of two studies are consistent with each other is a matter of degree and binary failure/success inferences are overly simplistic. Additionally, a failure/success inference based on only consideration of the quantitative results of a study ignores other important factors such as study design. For example, the results of a replication study by be markedly different to those of an original study, and thus be classified as a 'failed replication' - but if a difference in research design (such as the use of a more reliable measurement method in the replication) could explain the difference in results, then where is the failure? Both results may be reliable, they just have different theoretical explanations. And finally, why is it interesting to compare the findings of two studies rather than all relevant studies exploring the research question of interest (e.g., via meta-analysis)? In short, I think the whole notion of 'replication failure/success' needs more justification.

- The introduction should ideally provide a definition of replication as definitions can diverge widely (cf. Schmidt, 2009; Nosek & Errington, 2020).

- Further justification for the inclusion of meta-analysis as a measure of replication success would be helpful. It seems to me that the other measures involve a comparison of the findings from an original study and replication study, whereas meta-analysis involves a synthesis of the evidence from both studies – isn't that a fundamentally different inference?

Minor comments

- Regarding this point: "Publication bias has probably persisted because journals mostly seem to accept studies that are novel (i.e., they prefer original studies over replications; e.g., Neuliep & Crandall, 1993, p. 21), good (i.e., "clear, coherent, engaging, exciting;" Kerr, 1998, p. 204), and statistically significant (usually, $p < .05$; e.g., Maxwell, 1981, p. 17)." – perhaps a bit more nuance is needed as publication bias probably arises from a combination of journals' publication criteria and whether authors decide to submit findings or not. These are naturally interacting, but I don't think the latter point can be ignored as there are now many journals that accept research regardless of its novelty and significance (e.g., the PLOS journals).

- I enjoyed looking through the (nicely commented) analysis code. One suggestion - reproducibility of analysis code can be undermined by a reliance on particular software dependencies (packages, language versions, operating systems etc.) which can quickly become outdated (or just differ by computer). The reproducibility of the provided code could therefore be enhanced by sharing the software environment in which the code runs smoothly. Multiple options are available which vary in technical involvement e.g., Docker, Binder, Code Ocean. The latter is relatively user friendly.

Decision letter (RSOS-201697.R0)

Dear Ms Muradchianian

The Editors assigned to your paper RSOS-201697 "How Best to Quantify Replication Success? A Simulation Study on the Comparison of Replication Success Metrics" have now received comments from reviewers and would like you to revise the paper in accordance with the reviewer comments and any comments from the Editors. Please note this decision does not guarantee eventual acceptance.

Please submit your revised manuscript and required files (see below) no later than 21 days from today's (ie 12-Jan-2021) date. Note: the ScholarOne system will 'lock' if submission of the revision is attempted 21 or more days after the deadline. If you do not think you will be able to meet this deadline please contact the editorial office immediately.

on behalf of Prof Essi Viding (Subject Editor)
openscience@royalsociety.org

Associate Editor Comments to Author:

The comments of the reviewers suggest your paper needs some modification before it can be considered publishable; however, the in-depth feedback should allow you to submit a much improved manuscript as a revision. Please pay close attention to the comments of the reviewers to ensure that you fully address their comments, and that you also include a point-by-point response to them when you supply your revision. Good luck!

Reviewer comments to Author:

Reviewer: 1
Comments to the Author(s)
See reviewer notes.

Reviewer: 2
Comments to the Author(s)
Summary

This article reports a simulation study designed to compare different measures of 'replication success'. The manuscript is written clearly and concisely. The study appears technically sound; however, I would like to see a more detailed justification for various conceptual tenants of the project (see 'major comments' below). The study has commendable adherence to transparency – all data and analysis code have been made available. The study's conclusions appear well calibrated to the evidence and relevant limitations are appropriately acknowledged and discussed.

Major comments

- I felt the introduction could do more to justify why replication success is an interesting concept to explore. One could argue, for example, that the extent to which the findings of two studies are

consistent with each other is a matter of degree and binary failure/success inferences are overly simplistic. Additionally, a failure/success inference based on only consideration of the quantitative results of a study ignores other important factors such as study design. For example, the results of a replication study may be markedly different to those of an original study, and thus be classified as a 'failed replication' – but if a difference in research design (such as the use of a more reliable measurement method in the replication) could explain the difference in results, then where is the failure? Both results may be reliable, they just have different theoretical explanations. And finally, why is it interesting to compare the findings of two studies rather than all relevant studies exploring the research question of interest (e.g., via meta-analysis)? In short, I think the whole notion of 'replication failure/success' needs more justification.

- The introduction should ideally provide a definition of replication as definitions can diverge widely (cf. Schmidt, 2009; Nosek & Errington, 2020).

- Further justification for the inclusion of meta-analysis as a measure of replication success would be helpful. It seems to me that the other measures involve a comparison of the findings from an original study and replication study, whereas meta-analysis involves a synthesis of the evidence from both studies – isn't that a fundamentally different inference?

Minor comments

- Regarding this point: "Publication bias has probably persisted because journals mostly seem to accept studies that are novel (i.e., they prefer original studies over replications; e.g., Neuliep & Crandall, 1993, p. 21), good (i.e., "clear, coherent, engaging, exciting;" Kerr, 1998, p. 204), and statistically significant (usually, $p < .05$; e.g., Maxwell, 1981, p. 17)." – perhaps a bit more nuance is needed as publication bias probably arises from a combination of journals' publication criteria and whether authors decide to submit findings or not. These are naturally interacting, but I don't think the latter point can be ignored as there are now many journals that accept research regardless of its novelty and significance (e.g., the PLOS journals).

- I enjoyed looking through the (nicely commented) analysis code. One suggestion - reproducibility of analysis code can be undermined by a reliance on particular software dependencies (packages, language versions, operating systems etc.) which can quickly become outdated (or just differ by computer). The reproducibility of the provided code could therefore be enhanced by sharing the software environment in which the code runs smoothly. Multiple options are available which vary in technical involvement e.g., Docker, Binder, Code Ocean. The latter is relatively user friendly.

===PREPARING YOUR MANUSCRIPT===

Please ensure that you include an acknowledgements' section before your reference list/bibliography. This should acknowledge anyone who assisted with your work, but does not

qualify as an author per the guidelines at <https://royalsociety.org/journals/ethics-policies/openness/>.

===PREPARING YOUR REVISION IN SCHOLARONE===

-- Ensure that your data access statement meets the requirements at <https://royalsociety.org/journals/authors/author-guidelines/#data>. You should ensure that you cite the dataset in your reference list. If you have deposited data etc in the Dryad repository, please include both the 'For publication' link and 'For review' link at this stage.

-- If you have uploaded ESM files, please ensure you follow the guidance at <https://royalsociety.org/journals/authors/author-guidelines/#supplementary-material> to include a suitable title and informative caption. An example of appropriate titling and captioning may be found at https://figshare.com/articles/Table_S2_from_Is_there_a_trade-off_between_peak_performance_and_performance_breadth_across_temperatures_for_aerobic_scooping_in_teleost_fishes_/3843624.

Author's Response to Decision Letter for (RSOS-201697.R0)

See Appendix B.

RSOS-201697.R1 (Revision)

Review form: Reviewer 1 (Peder Isager)

Is the manuscript scientifically sound in its present form?

Yes

Are the interpretations and conclusions justified by the results?

No

Is the language acceptable?

Yes

Do you have any ethical concerns with this paper?

No

Have you any concerns about statistical analyses in this paper?

Yes

Recommendation?

Accept with minor revision (please list in comments)

Comments to the Author(s)

See attached word document for comments (Appendix C).

Review form: Reviewer 2

Is the manuscript scientifically sound in its present form?

Yes

Are the interpretations and conclusions justified by the results?

Yes

Is the language acceptable?

Yes

Do you have any ethical concerns with this paper?

No

Have you any concerns about statistical analyses in this paper?

No

Recommendation?

Accept with minor revision (please list in comments)

Comments to the Author(s)

Summary

Thank you for the opportunity to review a revision of this manuscript. I am grateful to the authors for their attention to my comments and changes made to the manuscript to address them. I continue to find much to like about the manuscript and as I stated in my first review, the substantive design and analysis appear technically sound and the reporting is clear and transparent. I have two remaining comments.

Specific comments

1. Previously I suggested that further justification for the inclusion of meta-analysis as a measure of replication success would be helpful because the other measures involve a binary comparison of the findings from an original study and replication study, whereas meta-analysis involves a synthesis of the evidence from both studies – to me that seems to be a fundamentally different inference. In their response, the authors justify this by saying they are just using measures that other projects have considered as indicators of replication success and are withholding their own judgement. In this case, they say, OSC used fixed-effect meta-analyses to conclude that “the number of successful replications was 68%”. However, I’m not sure that this is correct – in OSC they state that “combining original and replication results left 68% with statistically significant effects” - that is not the same as saying that 68% of replications were successful. Perhaps that needs adjusting, and I still think justification is needed for the inclusion of meta-analysis as a measure of replication success. Indeed, I also argued that a binary replication success/failure decision based on two studies is a rather impoverished inference relative what could be learned from a meta-analysis of all available evidence – a point which is now included in the discussion. So there’s at the very least some inconsistency here that perhaps needs addressing.

2. Minor point - it’s great that the authors created a Code Ocean container to enhance the reproducibility of the simulations – though don’t think a link was included in manuscript.

Anyhow, I hope the authors found it a relatively straightforward and useful process – thanks for giving it a shot!

Decision letter (RSOS-201697.R1)

Dear Ms Muradchianian,

On behalf of the Editors, we are pleased to inform you that your Manuscript RSOS-201697.R1 "How Best to Quantify Replication Success? A Simulation Study on the Comparison of Replication Success Metrics" has been accepted for publication in Royal Society Open Science subject to minor revision in accordance with the referees' reports. Please find the referees' comments along with any feedback from the Editors below my signature.

Please submit your revised manuscript and required files (see below) no later than 7 days from today's (ie 19-Apr-2021) date. Note: the ScholarOne system will 'lock' if submission of the revision is attempted 7 or more days after the deadline. If you do not think you will be able to meet this deadline please contact the editorial office immediately.

on behalf of Professor Essi Viding (Subject Editor)
openscience@royalsociety.org

Associate Editor Comments to Author:

Thank you for this revision, which the reviewers have kindly commented on for a second time. There are a number of remaining queries that need addressing - in particular, the comments included in the attachment: please carefully consider the best approach to these comments, and

ensure that you address them in your revision. Assuming the editors consider the changes to be sufficient and the responses satisfactory, your paper will be accepted.

Reviewer comments to Author:

Reviewer: 1

Comments to the Author(s)

See attached word document for comments.

Reviewer: 2

Comments to the Author(s)

Summary

Thank you for the opportunity to review a revision of this manuscript. I am grateful to the authors for their attention to my comments and changes made to the manuscript to address them. I continue to find much to like about the manuscript and as I stated in my first review, the substantive design and analysis appear technically sound and the reporting is clear and transparent. I have two remaining comments.

Specific comments

1. Previously I suggested that further justification for the inclusion of meta-analysis as a measure of replication success would be helpful because the other measures involve a binary comparison of the findings from an original study and replication study, whereas meta-analysis involves a synthesis of the evidence from both studies – to me that seems to be a fundamentally different inference. In their response, the authors justify this by saying they are just using measures that other projects have considered as indicators of replication success and are withholding their own judgement. In this case, they say, OSC used fixed-effect meta-analyses to conclude that “the number of successful replications was 68%”. However, I’m not sure that this is correct – in OSC they state that “combining original and replication results left 68% with statistically significant effects” - that is not the same as saying that 68% of replications were successful. Perhaps that needs adjusting, and I still think justification is needed for the inclusion of meta-analysis as a measure of replication success. Indeed, I also argued that a binary replication success/failure decision based on two studies is a rather impoverished inference relative what could be learned from a meta-analysis of all available evidence – a point which is now included in the discussion. So there’s at the very least some inconsistency here that perhaps needs addressing.

2. Minor point - it’s great that the authors created a Code Ocean container to enhance the reproducibility of the simulations – though don’t think a link was included in manuscript. Anyhow, I hope the authors found it a relatively straightforward and useful process – thanks for giving it a shot!

===PREPARING YOUR MANUSCRIPT===

===PREPARING YOUR REVISION IN SCHOLARONE===

- If you are providing image files for potential cover images, please upload these at this step, and inform the editorial office you have done so. You must hold the copyright to any image provided.
- A copy of your point-by-point response to referees and Editors. This will expedite the preparation of your proof.

- Ensure that your data access statement meets the requirements at <https://royalsociety.org/journals/authors/author-guidelines/#data>. You should ensure that you cite the dataset in your reference list. If you have deposited data etc in the Dryad repository, please only include the 'For publication' link at this stage. You should remove the 'For review' link.
- If you are requesting an article processing charge waiver, you must select the relevant waiver option (if requesting a discretionary waiver, the form should have been uploaded at Step 3 'File upload' above).
- If you have uploaded ESM files, please ensure you follow the guidance at <https://royalsociety.org/journals/authors/author-guidelines/#supplementary-material> to include a suitable title and informative caption. An example of appropriate titling and captioning may be found at https://figshare.com/articles/Table_S2_from_Is_there_a_trade-off_between_peak_performance_and_performance_breadth_across_temperatures_for_aerobic_scope_in_teleost_fishes_/3843624.

Author's Response to Decision Letter for (RSOS-201697.R1)

See Appendix D.

Decision letter (RSOS-201697.R2)

Dear Ms Muradchianian,

I am pleased to inform you that your manuscript entitled "How Best to Quantify Replication Success? A Simulation Study on the Comparison of Replication Success Metrics" is now accepted for publication in Royal Society Open Science.

You can expect to receive a proof of your article in the near future. Please contact the editorial office (openscience@royalsociety.org) and the production office (openscience_proofs@royalsociety.org) to let us know if you are likely to be away from e-mail contact – if you are going to be away, please nominate a co-author (if available) to manage the proofing process, and ensure they are copied into your email to the journal. Due to rapid publication and an extremely tight schedule, if comments are not received, your paper may experience a delay in publication.

on behalf of Professor Essi Viding (Subject Editor)
openscience@royalsociety.org

Appendix A

Article summary

This article investigates the relative performance of several metrics designed to indicate replication success. Diverging from previous work, this article defines replication success in terms of whether the metric leads to correct beliefs about the true population effect size being studied. A metric is successful if its result leads us to believe that the effect is >0 when it really is >0 (true positive), and if it leads us to believe that the effect is 0 when it really is 0 (true negative). The article investigates replication success by simulating 20000*2 effect sizes from one distribution with mean >0 and one distribution with mean 0. From these effect sizes, samples are randomly drawn that vary in their sample size. In addition, publication bias is simulated by varying the probability that samples with non-significant results are included in the analyses. The authors study the percentage of times that different methods lead to concluding that the underlying effect is true in both zero-effect (false positive) and non-zero effect (true positive) conditions. The main findings are (1) that the Small Telescopes and skeptical p-value approach perform qualitatively worse than all other metrics (either generating too few true positives or too many false positives), (2) that all other metrics perform at a similar level, that Bayesian metrics appear to slightly outperform non-Bayesian metrics, and (3) that metrics based on meta-analysis slightly outperform single study metrics under less than extreme publication bias but underperform relative to single study metrics under extreme publication bias.

Overall impression

Generally, my main impression of this article is that it is quite difficult to understand. There are several reasons for this I think, most of which has to do with the structure and writing of the article. In other words, my lack of understanding does not mean that I believe the simulation study is of low quality or that the substantive analyses are weak. Far from it. I think there is merit to this study design, and I think it in principle would be a worthwhile contribution to the scientific record. I simply find it difficult to assess the overall quality of the study design and analyses.

There are several independent reasons why I found the article hard to parse, which makes it difficult to nail this criticism into a few succinct and actionable comments. I am fully aware that such overall impressions are vague and hard for authors to address in a satisfying way. In my comments I have therefore tried to identify concrete issues that hampered my understanding while reading. These comments should be considered as examples of a more general problem, however, and I urge the authors to carefully consider whether changes could be made in each section to enhance readability.

Note also that my general confusion may have led me to make comments that are based on misunderstanding the purpose or procedures of the study. If such is the case, the authors should of course ignore my substantive comments, but I urge them to consider whether any aspect of their writing could be improved to protect future readers from similar misunderstandings.

Major comments

- 1) In the section that summarize the different metrics that will be compared in this study, I miss an explanation of how each of those methods relates to the definition of replication success. I.e. how would one use these metrics to make a decision that can either lead to a false or true positive statement about the underlying population effect? For example, how does one make decision about the population effect based on the Small Telescopes approach?

- 2) In general, I found the methods section difficult to understand. I will try to highlight what I think are the main reasons for my lack of understanding.
 - a. I miss a summary table of the parameters included in the simulation. Particularly those that are being varied. See table 1 in Smaldino et al. (2019, <https://doi.org/10.1098/rsos.190194>) for an example of the kind of table I am thinking of.
 - b. The procedure for generating samples from the zero- and non-zero populations confused me. Why not simply define two sets of population parameters and draw samples directly from these? Why sample a number of effect sizes first, then generate 20000 mini populations, and then draw sample from these? I appreciate that the two methods might be mathematically equivalent, but the roundabout method of sampling effect sizes in a separate step was difficult to wrap my head around.
 - c. Similarly, it was difficult for me to understand how publication bias was simulated and what consequences it had for sample size. Note that I am not implying that anything is wrong with this method of simulating publication bias. It was simply difficult to grasp. Perhaps a figure or diagram that visualizes the simulation process could be added for readability?
 - d.
- 3) No analysis seems to be reported of how results behave over variations in sample size. That seems peculiar to me, given that sample size is a parameter that is systematically varied. I recommend adding a summary of variations in results across sample size. Even if there are non-existent or negligible, this would be interesting to know.
- 4) Figure 2 – Figure 6: I find it very hard to parse these plots. I will try to summarize what I think are the main obstacles to my understanding:
 - a. First and foremost, I am simply unfamiliar with interpreting ROC plots.
 - b. The plots are split and spread out across several pages, but it seems relevant to be able to compare the different publication bias conditions. Is it possible to include all plots as panels in one larger plot so that conditions can be compared with the naked eye?
 - c. The x-axis does not represent the parameter that is varied – i.e. threshold values – and this takes a lot of time to get used to. In general, it is difficult to understand how threshold values vary across the plots. The symbols used as markers on the different lines are not very intuitive.
 - d. In the text, elements of the plots are sometimes referenced (e.g. lines crossing at certain values of the x-axis) that are very hard to spot in the plots due to the lines being so closely situated.
 - e. In general, the plots are quite information dense.
- 5) I might be misunderstanding the analysis setup, but I believe that the procedure for sampling studies from the zero-effect population is formally incorrect. The procedure appears to begin by

creating a distribution of population effect sizes. Then a sample of 20 000 effect sizes from this population is drawn (in R: $\mu_0 = \text{rnorm}(20000, 0, 0.02)$). Then one population distribution is created for each of the effect sizes sampled. Then one sample from each of these 20 000 population distributions is drawn (in R: $\text{out}_0 = \text{rnorm}(n, \mu_0, \sigma)$). If this is correct, then the procedure is not sampling 20 000 samples from a population of effect size zero. It is sampling 20 000 samples from 20 000 populations, all of which has a *true non-zero effect size*, albeit very small. In other words, if I am correct then the zero-effect condition does not represent a sample of studies drawn from populations with an effect size of zero. I suspect correcting this error would not lead to any substantive changes in the overall conclusions, but it should nonetheless be corrected.

Line-by-line comments

P4. L22-26: I miss a brief elaboration on what this discrepancy entails.

P5. L9-10: Shouldn't the false failure rate be defined as incompatible conclusions when their underlying true effect sizes do *not* differ?

P6. L7-11: The change from "correctly" to "incorrectly" between (1) and (2) is slightly confusing. Consider revising (2) to "methods correctly conclude there is no effect when there is none".

Table 1. 1) Significance: Why does it matter whether the effect size is positive? Isn't the important criterion that the effect is in the same direction in both replication and original study? This question also goes for 3), 4) and 6).

Table 1. 2) Small telescopes: The structure of the sentence makes the definition very hard to parse, and I believe the first "that" should be "than".

P9. L6-12: As it stands, the rationale provided here is not clear to me. The goal of this paragraph seems to be to explain to me what the significance criterion consists of. It does not seem like the right place to explain how the criterion is implemented in the simulation study. For example, I do not at this point understand what 100% publication bias means, nor do I understand the point of dropping the significant-in-original-study criterion for the purposes of simulation.

P9. L14-24: From this explanation it is unclear to me why the small telescopes approach has anything to do with replication success. I.e. I do not understand the rationale of the approach from this paragraph.

P11. L14-23: I do not see the justification for always doubling the sample size in the simulated replications. Is this attempt at model realism necessary, and is biasing the model in this way really worth it? Intuitively, I feel like it would be useful to how the model results behave under conditions where replication sample size is the same as, or lower than, the original sample size. Is this not true?

P12. L40-54: Is this trying to tell me that I should read every value X in table 2 as "X +/- CI"? If so, would it be possible to enhance readability by including the intervals directly in table 2?

P16. L32-36: I find it very difficult to spot where in figure 2 this supposed crossing takes place.

P16. L57-P22: L19: Is a paragraph for each plot really necessary? Could this section not simply be condensed into “difference in true/false positive rates remained consistent across all levels of publication bias below 100%”?

P23. L17-23: Would it be possible to give a quick summary of Anderson & Maxwell’s approach here? Otherwise this point is not very insightful, since I don’t understand what is meant by “theoretical considerations”.

P24. L19-26: Given the limitations mentioned above, I think it is reasonable and important to consider this a “first attempt” at comparing measures of replication success, and I sincerely appreciate this careful conclusion to the article. Perhaps this language could be added to the abstract as well?

Appendix B

March 12, 2021

Dear Dr. Viding,

We are submitting the revised manuscript “How Best to Quantify Replication Success? A Simulation Study on the Comparison of Replication Success Metrics” (RSOS-201697), for publication in Royal Society Open Science. Below is a detailed description of the ways in which we have incorporated your suggestions and those of the reviewers. We reproduced the relevant sections of the reviews in standard font and added our response in bold. We hope that this revision is satisfactory. We look forward to your comments.

Kind regards,

Jasmine Muradchianian

Rink Hoekstra

Henk Kiers

Don van Ravenzwaaij

Editor

The comments of the reviewers suggest your paper needs some modification before it can be considered publishable; however, the in-depth feedback should allow you to submit a much improved manuscript as a revision. Please pay close attention to the comments of the reviewers to ensure that you fully address their comments, and that you also include a point-by-point response to them when you supply your revision. Good luck!

E1: The reviewers provided us with great comments, and we did our best to fully address them. Additionally, we have made some changes regarding the metric sceptical p -value. When our manuscript was being reviewed, we had received comments from the developer of this method about some improvements they had made to the sceptical p -value. To be more specific, they gave us the advice to use “level=golden”, which is now the default option in their R package “ReplicationSuccess”, whereas in our previous version we had used “level=nominal”, which was the default option in their initial R package. By doing this, our results for the sceptical p -value are now different compared to our initial manuscript.

Reviewer 1

Article summary:

This article investigates the relative performance of several metrics designed to indicate replication success. Diverging from previous work, this article defines replication success in terms of whether the metric leads to correct beliefs about the true population effect size being studied. A metric is successful if its result leads us to believe that the effect is >0 when it really is >0 (true positive), and if it leads us to believe that the effect is 0 when it really is 0 (true negative). The article investigates replication success by simulating 20000*2 effect sizes from one distribution with mean >0 and one distribution with mean 0. From these effect sizes, samples are randomly drawn that vary in their sample size. In addition, publication bias is simulated by varying the probability that samples with non-significant results are included in the analyses. The authors study the percentage of times that different methods lead to concluding that the underlying effect is true in both zero-effect (false positive) and non-zero effect (true positive) conditions. The main findings are (1) that the Small Telescopes and skeptical p-value approach perform qualitatively worse than all other metrics (either generating too few true positives or too many false positives), (2) that all other metrics perform at a similar level, that Bayesian metrics appear to slightly outperform non-Bayesian metrics, and (3) that metrics based on meta-analysis slightly outperform single study metrics under less than extreme publication bias but underperform relative to single study metrics under extreme publication bias.

Overall impression:

Generally, my main impression of this article is that it is quite difficult to understand. There are several reasons for this I think, most of which has to do with the structure and writing of the article. In other words, my lack of understanding does not mean that I believe the simulation study is of low quality or that the substantive analyses are weak. Far from it. I think there is merit to this study design, and I think it in principle would be a worthwhile contribution to the scientific record. I simply find it difficult to assess the overall quality of the study design and analyses.

There are several independent reasons why I found the article hard to parse, which makes it difficult to nail this criticism into a few succinct and actionable comments. I am fully aware that such overall impressions are vague and hard for authors to address in a satisfying way. In my comments I have therefore tried to identify concrete issues that hampered my understanding while reading. These comments should be considered as examples of a more general problem, however, and I urge the authors to carefully consider whether changes could be made in each section to enhance readability.

R1: We are sorry to hear that the manuscript is quite difficult to understand. Therefore, we have read our manuscript carefully several times, and where we saw possibilities to enhance readability, we did so.

Note also that my general confusion may have led me to make comments that are based on misunderstanding the purpose or procedures of the study. If such is the case, the authors should of course ignore my substantive comments, but I urge them to consider whether any aspect of their writing could be improved to protect future readers from similar misunderstandings.

Major comments:

1) In the section that summarize the different metrics that will be compared in this study, I miss an explanation of how each of those methods relates to the definition of replication success. I.e. how would one use these metrics to make a decision that can either lead to a false or true positive statement about the underlying population effect? For example, how does one make decision about the population effect based on the Small Telescopes approach?

R1-1: Thank you for pointing this out. We have now included the following information in the first paragraph of the methods section:

“We mostly follow replication studies that use these measures in order to quantify replication success. Specifically, we follow the procedures applied by large replication projects such as Camerer and colleagues (5, 6), Cova and colleagues (9), and OSC (27). In these studies, replication success is usually operationalized as a positive result in the replication attempt following a positive result in the original study (original null results are replicated less often, since there are far fewer original null results in the scientific literature). In a sense, the original study result is the golden standard against which to anchor the replication. In a simulation study, one has the advantage of knowing the true state of the world. Rather than using the original sample result as a proxy for the true population effect, one can evaluate the replication attempt against the ‘known truth’ directly. For this reason, in addition to assessing replication success, we go one step further namely, identifying true and false positives. In the context of our study, a true positive refers to obtaining a positive replication result when the underlying true effect is non-zero; a false positive refers to obtaining a positive replication result when the underlying true effect is practically zero.”

Additionally, we have now included Table 3 in our methods section, where we provide an example of a true and false positive finding for each metric.

2) In general, I found the methods section difficult to understand. I will try to highlight what I think are the main reasons for my lack of understanding.

- a. I miss a summary table of the parameters included in the simulation. Particularly those that are being varied. See table 1 in Smaldino et al. (2019, <https://doi.org/10.1098/rsos.190194>) for an example of the kind of table I am thinking of.

R1-2a: Thank you for the suggestion. We added the following table to our manuscript as Table 2:

Table 2

Summary of Parameter Values Included in the Simulation

Parameter	Definition	Simulated values
iter	Number of iterations per simulation cell	5000
n	sample size original studies	25, 50, 75, 100
n.rep	sample size replication studies	50, 100, 150, 200
truth	true population effect size distribution	~N(mean=0, SD=0.02) ~N(mean=0.5, SD=0.15)
publication bias	percentage of non-significant findings that are published relative to the percentage of significant findings that are published	100, 75, 50, 25, 0

- b. The procedure for generating samples from the zero- and non-zero populations confused me. Why not simply define two sets of population parameters and draw samples directly from these? Why sample a number of effect sizes first, then generate 20000 mini populations, and then draw sample from these? I appreciate that the two methods might be mathematically equivalent, but the roundabout method of sampling effect sizes in a separate step was difficult to wrap my head around.

R1-2b: Perhaps to clarify, we do ‘define two sets of population parameters and draw samples directly from these’. Our two sets of population parameters are defined as $N(0, 0.02)$ and $N(0.5, 0.15)$, respectively. The rationale behind the decision of randomly drawing 20,000 standardized population effect sizes from each distribution, rather than using fixed underlying true effect sizes, is to obtain variation across the true population effect sizes. We do this in order to stay as close as possible, in our opinion, to the reality. Therefore, we introduce some variation across the effect sizes. We apologize for the confusion on this front and hope that the inserted Table 2 helps to clarify this (see previous answer). Additionally, we have included this rationale in the methods section of our revised manuscript (see subsection “Underlying true population effects”).

- c. Similarly, it was difficult for me to understand how publication bias was simulated and what consequences it had for sample size. Note that I am not implying that anything is wrong with this method of simulating publication bias. It was simply difficult to grasp. Perhaps a figure or diagram that visualizes the simulation process could be added for readability?

R1-2c: We did not check what consequences publication bias had for sample size, because this is beyond the scope of our study. In order to clarify the publication process, we now included Figure 1 in our manuscript.

d.

3) No analysis seems to be reported of how results behave over variations in sample size. That seems peculiar to me, given that sample size is a parameter that is systematically varied. I recommend adding a summary of variations in results across sample size. Even if there are nonexistent or negligible, this would be interesting to know.

R1-3: Although your suggestion is very interesting, we decided not to look at how results behave over variation in sample size, because this is beyond the scope of our study. We did vary the original and replication sample sizes in order to obtain a more realistic scenario, since in daily science, the sample sizes of studies do vary. Therefore, we lump everything together in order to make our created reality more realistic. In addition, we prefer not to add an additional layer of complexity to the simulation results. If, for instance, we would wish to do a systematic comparison across five different sample sizes for the existing cells in the simulation, we would wind up with 25 instead of 5 figures with simulation results. In our opinion, the current paper is already dense in terms of numerical results and we do not think it is wise to compound the issue.

4) Figure 2 – Figure 6: I find it very hard to parse these plots. I will try to summarize what I think are the main obstacles to my understanding:

- a. First and foremost, I am simply unfamiliar with interpreting ROC plots.

R1-4a: We realize that many of the intended readers may not be familiar with ROC plots as well. Nevertheless, we think explaining them in more detail would, in our opinion, distract from the core of our paper.

- b. The plots are split and spread out across several pages, but it seems relevant to be able to compare the different publication bias conditions. Is it possible to include all plots as panels in one larger plot so that conditions can be compared with the naked eye?

R1-4b: We agree that having the plots side by side would facilitate direct comparisons and we attempted to put all figures as multiple panels in one plot in the submission stage. Unfortunately, we found that this reduces the size of the plots to the point where they are very difficult to read. If our paper should be accepted, perhaps the type setter can do something to facilitate a comparison across figures at the production stage.

- c. The x-axis does not represent the parameter that is varied – i.e. threshold values – and this takes a lot of time to get used to. In general, it is difficult to understand how threshold values vary across the plots. The symbols used as markers on the different lines are not very intuitive.

R1-4c: We have opted for ROC curves, because the focus of our study is not only on the threshold values. When putting threshold values on the x-axis, we can compare an alpha level of .05 to an alpha level of .10 *within* the significance method. We cannot make comparisons *across* methods though (should we compare an alpha of .05 to a Bayes factor threshold of 3? Or a Bayes factor threshold of 2? Or...). The ROC curves allow one to examine whether for each threshold level of one method (e.g., significance testing), there exists a threshold level for another method (e.g., Bayesian testing) that is superior in terms of both true and false positive rates.

- d. In the text, elements of the plots are sometimes referenced (e.g. lines crossing at certain values of the x-axis) that are very hard to spot in the plots due to the lines being so closely situated.

R1-4d: We have thought long and hard how to best deal with this in the context of the first ROC plot (lines in the remaining ROC plots are more clearly differentiable in our opinion). We opted to provide some guidance on a cross-over pattern that is hard to see by eye for the first ROC plot. The lines are closely situated because the differences between particular metrics are very small. We welcome concrete suggestions how to improve our plots, of course.

- e. In general, the plots are quite information dense.

R1-4e: We agree with this observation. A different choice could be to further subdivide the data (i.e., to split the results up into 10 plots, or even 20 plots). We believe this costs more than it gains (i.e., additional figures take lots of extra space, it becomes harder to compare results across simulation cells). We prefer to keep it as is for these reasons, but are happy to defer to the editor on this.

5) I might be misunderstanding the analysis setup, but I believe that the procedure for sampling studies from the zero-effect population is formally incorrect. The procedure appears to begin by creating a distribution of population effect sizes. Then a sample of 20 000 effect sizes from this population is drawn (in R: $\mu_0 = rnorm(20000, 0, 0.02)$). Then one population distribution is created for each of the effect sizes sampled. Then one sample from each of these 20 000 population distributions is drawn (in R: $out_0 = rnorm(n, \mu_0, \sigma)$). If this is correct, then the procedure is not sampling 20 000 samples from a population of effect size zero. It is sampling 20 000 samples from 20 000 populations, all of which has a true non-zero effect size, albeit very small. In other words, if I am correct then the zero-effect condition does not represent a sample of studies drawn from populations with an effect size of zero. I suspect correcting this error would not lead to any substantive changes in the overall conclusions, but it should nonetheless be corrected.

R1-5: Choosing for effect sizes very close to, but not exactly, zero was a conscious choice. One could argue that it is hard to imagine that in actual practice, samples are drawn from populations with mean exactly equal to 0. It seems more realistic and important to us to try to distinguish populations with means very close to zero from those with means considerably larger than 0. In the manuscript we clarify this now in the methods section (subsection “Underlying true population effects”).

Line-by-line comments

P4. L22-26: I miss a brief elaboration on what this discrepancy entails.

R1-P4. L22-26: Thank you for noticing this. We changed this part such that the current description is clearer and more relevant in our opinion. We have included the following information in our introduction:

“The focus of the present study is on close replications, which “aim to recreate a study as closely as possible, so that ideally the only differences between the two are the inevitable ones (e.g., different participants)” (3, p. 218). Rather than exact or direct replications, we prefer the term “close replications” because it highlights that “there is no such thing as an exact replication” (33, p. 92). In the present study, we make an attempt to answer “... the question of how to judge and quantify replication success” (40, p. 1457).”

P5. L9-10: Shouldn't the false failure rate be defined as incompatible conclusions when their underlying true effect sizes do not differ?

R1-P5. L9-10: In order to prevent any confusion, we now provide the exact definition such as used by Schauer and Hedges (2020) in our introduction. We included the following information:

“In their study, they analytically assess two types of classification errors: 1) false success rate, which is the proportion of times “the analysis concludes that the studies successfully replicated when they did not (for a given definition of replication)” (32, p. 3) over all times the analysis concludes that the studies successfully replicated, and 2) false failure rate, which is the proportion of times of an “analysis concluding that the studies failed to replicate when they actually successfully replicated according to a given definition” (32, p. 3) over all times the analysis concludes that the studies failed to replicate.”

P6. L7-11: The change from “correctly” to “incorrectly” between (1) and (2) is slightly confusing. Consider revising (2) to “methods correctly conclude there is no effect when there is none”.

R1-P6. L7-11: Unfortunately, the revised definition would be inaccurate, since the second category concerns incorrect/wrong/erratic conclusions. We will retain (2) methods incorrectly conclude there is an effect when there is not, but will slant the words correct, incorrect, one and not in the definitions for emphasis.

Table 1. 1) Significance: Why does it matter whether the effect size is positive? Isn't the important criterion that the effect is in the same direction in both replication and original study? This question also goes for 3), 4) and 6).

R1-Table 1. 1): The reason why we focus on positive effect sizes is because the one sample t test in each study is one-sided positive (i.e., $H_0: \mu = 0$ and $H_a: \mu > 0$). We did this for the sake of simplicity of the present study.

Table 1. 2) Small telescopes: The structure of the sentence makes the definition very hard to parse, and I believe the first “that” should be “than”.

R1-Table 1. 2): Thank you for spotting the typo; “that” should indeed be “than”. As for the description of the Small Telescopes method, we have done our best to describe a method that is comparatively involved. We believe that the relevant section in the methods section along with our description in Table 1 should provide the reader with enough handholds to understand the Small Telescopes method.

P9. L6-12: As it stands, the rationale provided here is not clear to me. The goal of this paragraph seems to be to explain to me what the significance criterion consists of. It does not seem like the right place to explain how the criterion is implemented in the simulation study. For example, I do not at this point understand what 100% publication bias means, nor do I understand the point of dropping the significant-in-original-study criterion for the purposes of simulation.

R1-P9. L6-12: Thank you for pointing this out. We have included the following information in our methods section:

“In this simulation study, we operationalize replication success for this method as a significant result in the replication study alone in order to be able to apply this method to original studies without a significant result as well.”

P9. L14-24: From this explanation it is unclear to me why the small telescopes approach has anything to do with replication success. I.e. I do not understand the rationale of the approach from this paragraph.

R1-10: We understand that this approach may sound complicated. It took us a fair bit of time and effort to apply this method in our study. However, since an important aspect of the present study was to evaluate all metrics that have been proposed for quantifying replication success, we thought we would be negligent if we would have left out the Small Telescopes method (see also our response to R2-3).

P11. L14-23: I do not see the justification for always doubling the sample size in the simulated replications. Is this attempt at model realism necessary, and is biasing the model in this way really worth it? Intuitively, I feel like it would be useful to how the model results behave under conditions where replication sample size is the same as, or lower than, the original sample size. Is this not true?

R1-P11. L14-23: An important goal in our study was to stay as close as possible to scientific practice. In replication studies, sample sizes are typically larger than the original sample sizes, which is also the case in OSC (27; in their “final” data file in R), Camerer and colleagues (5; Table S1 in Supplementary Materials on p. 19), Camerer and colleagues (6; Table 3 in Supplementary Materials on p. 54), and Cova and colleagues (9; on p. 8).

P12. L40-54: Is this trying to tell me that I should read every value X in table 2 as “ $X \pm CI$ ”? If so, would it be possible to enhance readability by including the intervals directly in table 2?

R1-P12. L40-54: The point of providing this information was to communicate that the uncertainty margins are very small. They are not exact values though, and including them in the table would give the impression that they are. As such, we prefer to provide this disclaimer in the text only. In addition, we believe the table to be more readable without intervals, rather than with.

P16. L32-36: I find it very difficult to spot where in figure 2 this supposed crossing takes place.

R1- P16. L32-36: Please see our previous answer R1-4d.

P16. L57-P22: L19: Is a paragraph for each plot really necessary? Could this section not simply be condensed into “difference in true/false positive rates remained consistent across all levels of publication bias below 100%”?

R1-P16. L57-P22: L19: We found the ROC plots quite information dense, so therefore, we decided to include one paragraph for each figure. In addition, since the behaviour of the sceptical p -value varies across the publication bias scenarios in our revised manuscript, we believe that including a paragraph for each plot might enhance readability.

P23. L17-23: Would it be possible to give a quick summary of Anderson & Maxwell’s approach here? Otherwise this point is not very insightful, since I don’t understand what is meant by “theoretical considerations”.

R1-P23. L17-23: Agreed, we have included the underlined information in our discussion:

“For example, Anderson and Maxwell (2016) suggest using theory and past research to construct a confidence interval (p. 5, 9). Since we conducted a simulation study, only metrics were included for which such theoretical considerations are not required, because the simulated ‘phenomena’ were basically numbers representing fictitious studies, and hence theory-free.”

P24. L19-26: Given the limitations mentioned above, I think it is reasonable and important to consider this a “first attempt” at comparing measures of replication success, and I sincerely appreciate this careful conclusion to the article. Perhaps this language could be added to the abstract as well?

R1- P24. L19-26: Agreed, we have included the underlined information:

“To overcome the frequently debated crisis of confidence, replicating studies is becoming increasingly more common. Multiple frequentist and Bayesian measures have been proposed to evaluate whether a replication is successful, but little is known about which method best captures replication success. This study is one of the first attempts to compare a number of quantitative measures of replication success with respect to their ability to draw the correct inference when the underlying truth is known, while taking publication bias into account. Our results show that Bayesian metrics seem to slightly outperform frequentist metrics across the board. Generally, meta-analytic approaches seem to slightly outperform metrics that evaluate single studies, except in the scenario of extreme publication bias, where this pattern reverses.”

Reviewer 2

Summary:

This article reports a simulation study designed to compare different measures of ‘replication success’. The manuscript is written clearly and concisely. The study appears technically sound; however, I would like to see a more detailed justification for various conceptual tenants of the project (see ‘major comments’ below). The study has commendable adherence to transparency – all data and analysis code have been made available. The study’s conclusions appear well calibrated to the evidence and relevant limitations are appropriately acknowledged and discussed.

Major comments:

- I felt the introduction could do more to justify why replication success is an interesting concept to explore. One could argue, for example, that the extent to which the findings of two studies are consistent with each other is a matter of degree and binary failure/success inferences are overly simplistic. Additionally, a failure/success inference based on only consideration of the quantitative results of a study ignores other important factors such as study design. For example, the results of a replication study by be markedly different to those of an original study, and thus be classified as a ‘failed replication’ – but if a difference in research design (such as the use of a more reliable measurement method in the replication) could explain the difference in results, then where is the failure? Both results may be reliable, they just have different theoretical explanations. And finally, why is it interesting to compare the findings of two studies rather than all relevant studies exploring the research question of interest (e.g., via meta-analysis)? In short, I think the whole notion of ‘replication failure/success’ needs more justification.

R2-1: Thank you very much for pointing this out. We mentioned these points in our discussion. In the introduction, we prefer to try to stay as close as possible to daily science, without including our judgement.

- The introduction should ideally provide a definition of replication as definitions can diverge widely (cf. Schmidt, 2009; Nosek & Errington, 2020).

R2-2: Agreed, we have included the following information in our introduction:

“The focus of the present study is on close replications, which “aim to recreate a study as closely as possible, so that ideally the only differences between the two are the inevitable ones (e.g., different participants)” (3, p. 218). Rather than exact or direct replications, we prefer the term “close replications” because it highlights that “there is no such thing as an exact replication” (33, p. 92).”

- Further justification for the inclusion of meta-analysis as a measure of replication success would be helpful. It seems to me that the other measures involve a comparison of the findings from an original study and replication study, whereas meta-analysis involves a synthesis of the evidence from both studies – isn't that a fundamentally different inference?

R2-3: We have tried to stay away from personal judgment of each of the methods in terms of suitability and focused mostly on the methods used in the big replication projects (5, 6, 9, 27). For example, using the traditional significance method, OSC (27) concluded that 36% of the replication studies in their sample were successful, whereas using fixed-effect meta-analyses, the number of successful replications was 68%. So here they also compare meta-analytic approaches to metrics that evaluate single studies (see also our response to R1-10).

Minor comments:

- Regarding this point: “Publication bias has probably persisted because journals mostly seem to accept studies that are novel (i.e., they prefer original studies over replications; e.g., Neuliep & Crandall, 1993, p. 21), good (i.e., “clear, coherent, engaging, exciting;” Kerr, 1998, p. 204), and statistically significant (usually, $p < .05$; e.g., Maxwell, 1981, p. 17).” – perhaps a bit more nuance is needed as publication bias probably arises from a combination of journals' publication criteria and whether authors decide to submit findings or not. These are naturally interacting, but I don't think the latter point can be ignored as there are now many journals that accept research regardless of its novelty and significance (e.g., the PLOS journals).

R2-4: Thank you very much for pointing this out. We have included the underlined information in our introduction:

“Publication bias has probably persisted because journals mostly seem to accept studies that are novel (i.e., they prefer original studies over replications; e.g., 25), good (i.e., “clear, coherent, engaging, exciting;” 18, p. 204), and statistically significant (usually, $p < .05$; e.g., 24). On the other hand, publication bias has probably also persisted because researchers are more likely to submit significant than nonsignificant results for publication (8). Given the fact that presenting significant results enhances the probability of a paper getting published (21), researchers often deviate from their original designs (by, for example, adding observations, dropping conditions, including control variables, etc.), sometimes without being aware that this artificially inflates Type I error rates (34).”

- I enjoyed looking through the (nicely commented) analysis code. One suggestion - reproducibility of analysis code can be undermined by a reliance on particular software dependencies (packages, language versions, operating systems etc.) which can quickly become outdated (or just differ by computer). The reproducibility of the provided code could therefore be enhanced by sharing the software environment in which the code runs smoothly. Multiple options are available which vary in technical involvement e.g., Docker, Binder, Code Ocean. The latter is relatively user friendly.

R2-4: Thank you very much for this advice. We uploaded the R codes and the R results to Code Ocean.

Appendix C

Comments to the authors

Overall, I feel that the vast majority of my round 1 concerns have been satisfyingly addressed by the authors. The addition of figure 1 and table 2 and 3, and the additional clarifications added to the manuscript, does much to improve readability. In the round 2 comments to authors, I will focus on the few points I think are worthwhile to follow up on one more time before the article is accepted for publication:

Major comment:

5) I might be misunderstanding the analysis setup, but I believe that the procedure for sampling studies from the zero-effect population is formally incorrect. The procedure appears to begin by creating a distribution of population effect sizes. Then a sample of 20 000 effect sizes from this population is drawn (in R: $\mu_0 = rnorm(20000, 0, 0.02)$). Then one population distribution is created for each of the effect sizes sampled. Then one sample from each of these 20 000 population distributions is drawn (in R: $out_0 = rnorm(n, \mu_0, \sigma)$). If this is correct, then the procedure is not sampling 20 000 samples from a population of effect size zero. It is sampling 20 000 samples from 20 000 populations, all of which has a true non-zero effect size, albeit very small. In other words, if I am correct then the zero-effect condition does not represent a sample of studies drawn from populations with an effect size of zero. I suspect correcting this error would not lead to any substantive changes in the overall conclusions, but it should nonetheless be corrected.

Author reply: *Choosing for effect sizes very close to, but not exactly, zero was a conscious choice.*

One could argue that it is hard to imagine that in actual practice, samples are drawn from populations with mean exactly equal to 0. It seems more realistic and important to us to try to distinguish populations with means very close to zero from those with means considerably larger than 0. In the manuscript we clarify this now in the methods section (subsection “Underlying true population effects”).

Reviewer reply to reply:

I appreciate that this decision is now at least made explicit in the methods section, and I can understand the authors’ desire to aim for model realism. However, I think the benefits of model realism is in this case outweighed by the conceptual confusion this move creates.

As an example, consider the following: all test procedures whose performance is evaluated in this manuscript is designed to test the hypothesis that the effect is exactly zero. This means that, with a high enough sample size, all procedures would *correctly* detect the presence of a small true effect in the “spurious effect” condition. Consequently, all procedures would appear to be performing very poorly because the performance criterion in this study is not being able to separate >0 from 0, but being able to separate practically meaningful from practically meaningless. However, the procedures tested here were never intended to evaluate whether the effects are meaningless, only whether effects are 0 or not, so they would only be performing poorly because they are not being used for what they were designed to do. If one wants procedures for detecting the presence of a practically meaningless effect, one should

simply use different methods, like equivalence testing, ROPE, or Bayes Factors where the null hypothesis includes non-zero values.

Additionally, it is not completely clear what terms such as “false positive”, “correct inference”, and “performance” means. Or, at least, the meaning of these terms in the present study is different from how the usual meaning ascribed to them.

Finally, there is no justification for the bound set on “spurious” in this simulation study, which seems to be $\mu \leq 0.02$. While the choice of bound is likely to be a somewhat arbitrary decision in a simulation study like this, the subjective nature of this parameter should at least be acknowledged and perhaps even listed in table 2.

I do not wish to be a stick in the mud on this point. If the authors feel that model realism is outweighs conceptual confusion here and simply wish to rebut my concern without making any analysis revisions, I think that could be acceptable so long as the manuscript more clearly emphasizes that all procedures in this study are being evaluated on a criterion that we *would like* them to perform well on (separating meaningful from spurious true effects) but not on a criterion they were necessarily *designed* to perform well on. One good place to bring up this issue seems to be on page 15, line 43-55.

Minor comments:

In “original studies” and “replication studies”, perhaps the data generation process could be described in a little more detail? I.e., it seems relevant to mention that data were randomly generated from a normal distribution centered on *truth*, with a constant standard deviation of 1. This is clear from the analysis R code, but feels like a central enough aspect of data generation that it ought to be included in the methods section.

Table 1. 1) Significance: Why does it matter whether the effect size is positive? Isn't the important criterion that the effect is in the same direction in both replication and original study? This question also goes for 3), 4) and 6).

Author reply: The reason why we focus on positive effect sizes is because the one sample t test in each study is one-sided positive (i.e., $H_0: \mu = 0$ and $H_a: \mu > 0$). We did this for the sake of simplicity of the present study.

Reviewer reply to reply: I take it then that table 1 is not trying to define replication success criteria for each metric *in general*, but more specifically define success criteria in the context of *the current simulation study*? Completely fair if so, but it might be wise to explicitly mention this in the manuscript.

P16. L32-36: I find it very difficult to spot where in figure 2 this supposed crossing takes place.

Author reply: Please see our previous answer R1-4d.

Reviewer reply to reply: Perhaps the (x,y) coordinates for where in the plot this crossing happens could be explicitly stated in the text? That way, the reader has a guide for where in the plot to look, and nothing about the plot itself needs to change.

Appendix D

April 23, 2021

Dear Dr. Viding,

We are submitting the revised manuscript “How Best to Quantify Replication Success? A Simulation Study on the Comparison of Replication Success Metrics” (RSOS-201697.R1), for publication in Royal Society Open Science. Below is a detailed description of the ways in which we have incorporated your suggestions and those of the reviewers. We reproduced the relevant sections of the reviews in standard font and added our response in bold. We hope that this revision is satisfactory. We look forward to your comments.

Kind regards,

Jasmine Muradchianian
Rink Hoekstra
Henk Kiers
Don van Ravenzwaaij

Editor

Thank you for this revision, which the reviewers have kindly commented on for a second time. There are a number of remaining queries that need addressing - in particular, the comments included in the attachment: please carefully consider the best approach to these comments, and ensure that you address them in your revision. Assuming the editors consider the changes to be sufficient and the responses satisfactory, your paper will be accepted.

E1: We would like to thank the reviewers for their time and effort in reading and commenting on our revised manuscript, and we did our best to fully address the comments in this second revision.

Reviewer 1

Comments to the authors:

Overall, I feel that the vast majority of my round 1 concerns have been satisfyingly addressed by the authors. The addition of figure 1 and table 2 and 3, and the additional clarifications added to the manuscript, does much to improve readability.

R1: We are very pleased to hear that the readability in our previous revision has improved.

In the round 2 comments to authors, I will focus on the few points I think are worthwhile to follow up on one more time before the article is accepted for publication:

Major comment:

5) I might be misunderstanding the analysis setup, but I believe that the procedure for sampling studies from the zero-effect population is formally incorrect. The procedure appears to begin by creating a distribution of population effect sizes. Then a sample of 20 000 effect sizes from this population is drawn (in R: $\mu_0 = \text{rnorm}(20000, 0, 0.02)$). Then one population distribution is created for each of the effect sizes sampled. Then one sample from each of these 20 000 population distributions is drawn (in R: $\text{out}_0 = \text{rnrom}(n, \mu_0, \text{sigma})$). If this is correct, then the procedure is not sampling 20 000 samples from a population of effect size zero. It is sampling 20 000 samples from 20 000 populations, all of which has a true non-zero effect size, albeit very small. In other words, if I am correct then the zero-effect condition does not represent a sample of studies drawn from populations with an effect size of zero. I suspect correcting this error would not lead to any substantive changes in the overall conclusions, but it should nonetheless be corrected.

Author reply: Choosing for effect sizes very close to, but not exactly, zero was a conscious choice.

One could argue that it is hard to imagine that in actual practice, samples are drawn from populations with mean exactly equal to 0. It seems more realistic and important to us to try to distinguish populations with means very close to zero from those with means considerably larger than 0. In the manuscript we clarify this now in the methods section (subsection “Underlying true population effects”).

Reviewer reply to reply:

I appreciate that this decision is now at least made explicit in the methods section, and I can understand the authors’ desire to aim for model realism. However, I think the benefits of model realism is in this case outweighed by the conceptual confusion this move creates. As an example, consider the following: all test procedures whose performance is evaluated in this manuscript is designed to test the hypothesis that the effect is exactly zero. This means that, with a high enough sample size, all procedures would correctly detect the presence of a small true effect in the “spurious effect” condition. Consequently, all procedures would appear to be performing very poorly because the performance criterion in this study is not being able to separate >0 from 0, but being able to separate practically meaningful from practically meaningless. However, the procedures tested here were never intended to evaluate whether the effects are meaningless, only whether effects are 0 or not, so they would only be performing poorly because they are not being used for what they were designed to do. If one wants procedures for detecting the presence of a practically meaningless effect, one should

simply use different methods, like equivalence testing, ROPE, or Bayes Factors where the null hypothesis includes non-zero values.

Additionally, it is not completely clear what terms such as “false positive”, “correct inference”, and “performance” means. Or, at least, the meaning of these terms in the present study is different from how the usual meaning ascribed to them.

Finally, there is no justification for the bound set on “spurious” in this simulation study, which seems to be $\mu \leq 0.02$. While the choice of bound is likely to be a somewhat arbitrary decision in a simulation study like this, the subjective nature of this parameter should at least be acknowledged and perhaps even listed in table 2.

I do not wish to be a stick in the mud on this point. If the authors feel that model realism is outweighs conceptual confusion here and simply wish to rebut my concern without making any analysis revisions, I think that could be acceptable so long as the manuscript more clearly emphasizes that all procedures in this study are being evaluated on a criterion that we would like them to perform well on (separating meaningful from spurious true effects) but not on a criterion they were necessarily designed to perform well on. One good place to bring up this issue seems to be on page 15, line 43-55.

R1-1: Thank you very much for pointing this out. We have now included the following information in the discussion section:

“Additionally, we would like to emphasize that all procedures in this study are being evaluated on a criterion that we would like them to perform well on (separating meaningful from spurious true effects), but not on a criterion they were necessarily designed to perform well on.”

Minor comments:

In “original studies” and “replication studies”, perhaps the data generation process could be described in a little more detail? I.e., it seems relevant to mention that data were randomly generated from a normal distribution centered on truth, with a constant standard deviation of 1. This is clear from the analysis R code, but feels like a central enough aspect of data generation that it ought to be included in the methods section.

R1-2: We think by including this information in our methods section might make the data generation process clearer to the reader, so therefore, we included the following information (subsection “Original studies”):

“The data were randomly generated from a normal distribution centered on truth, with a constant standard deviation of 1.”

Table 1. 1) Significance: Why does it matter whether the effect size is positive? Isn't the important criterion that the effect is in the same direction in both replication and original study? This question also goes for 3), 4) and 6).

Author reply: The reason why we focus on positive effect sizes is because the one sample t test in each study is one-sided positive (i.e., $H_0: \mu = 0$ and $H_a: \mu > 0$). We did this for the sake of simplicity of the present study.

Reviewer reply to reply: I take it then that table 1 is not trying to define replication success criteria for each metric in general, but more specifically define success criteria in the context

of the current simulation study? Completely fair if so, but it might be wise to explicitly mention this in the manuscript.

R1-3: Thank you very much for noticing this. We have now included the following underlined information in our method section (subsection “Replication success metrics”):

“In Table 1, the metrics of replication success and their associated criteria as operationalized in this study are summarized.”

P16. L32-36: I find it very difficult to spot where in figure 2 this supposed crossing takes place.

Author reply: Please see our previous answer R1-4d.

Reviewer reply to reply: Perhaps the (x,y) coordinates for where in the plot this crossing happens could be explicitly stated in the text? That way, the reader has a guide for where in the plot to look, and nothing about the plot itself needs to change.

R1-4: We now added this information in the results section when crossings happen.

Reviewer 2

Comments to the Author(s):

Summary:

Thank you for the opportunity to review a revision of this manuscript. I am grateful to the authors for their attention to my comments and changes made to the manuscript to address them. I continue to find much to like about the manuscript and as I stated in my first review, the substantive design and analysis appear technically sound and the reporting is clear and transparent. I have two remaining comments.

Specific comments:

1. Previously I suggested that further justification for the inclusion of meta-analysis as a measure of replication success would be helpful because the other measures involve a binary comparison of the findings from an original study and replication study, whereas meta-analysis involves a synthesis of the evidence from both studies – to me that seems to be a fundamentally different inference. In their response, the authors justify this by saying they are just using measures that other projects have considered as indicators of replication success and are withholding their own judgement. In this case, they say, OSC used fixed-effect meta-analyses to conclude that “the number of successful replications was 68%”. However, I’m not sure that this is correct – in OSC they state that “combining original and replication results left 68% with statistically significant effects”- that is not the same as saying that 68% of replications were successful. Perhaps that needs adjusting, and I still think justification is needed for the inclusion of meta-analysis as a measure of replication success. Indeed, I also argued that a binary replication success/failure decision based on two studies is a rather impoverished inference relative what could be learned from a meta-analysis of all available evidence – a point which is now included in the discussion. So there’s at the very least some inconsistency here that perhaps needs addressing.

R2-1: Thank you for pointing this out. We now included the following underlined information in our introduction:

“For example, using the traditional significance method, Open Science Collaboration (OSC, 27) concluded that 36% of the replication studies in their sample were successful, whereas when combining the original and replication effect sizes for cumulative evidence by using fixed-effects meta-analyses, the number of significant effects was 68%.”

Regarding the justification for the inclusion of meta-analysis as a measure of replication success, we agree that meta-analysis involves a synthesis of the evidence from both studies. However, we decided to choose a more pragmatic approach.

2. Minor point - it’s great that the authors created a Code Ocean container to enhance the reproducibility of the simulations – though don’t think a link was included in manuscript. Anyhow, I hope the authors found it a relatively straightforward and useful process – thanks for giving it a shot!

R2-2: Sorry for the confusion. This is the first time for us using Code Ocean. Initially, we thought that we used Code Ocean in a correct way. However, after trying to publish our R codes and the results on Code Ocean, we noticed that the standard possibilities in Code Ocean do not really seem to fit our study, since our first R code runs for days, while the amount of time Code Ocean provides seems to be much shorter. Because of this reason, we decided not to publish our study on Code Ocean. However, we would like

to thank you for mentioning this point, and we will take it into account when writing future articles.